# A Study of Two Impactful Heavy Rainfall Events in the Southern Appalachian Mountains during Early 2020, Part II; Regional Overview, Rainfall Evolution, and Satellite QPE Utility

Douglas Miller [1,*], Malarvizhi Arulraj [2], Ralph Ferraro [3], Christopher Grassotti [2], Bob Kuligowski [3], Shuyan Liu [4], Veljko Petkovic [2], Shaorong Wu [5] and Pingping Xie [5]

1   Atmospheric Sciences Department, University of North Carolina Asheville, Asheville, NC 28804, USA
2   ESSIC Cooperative Institute for Satellite Earth System Studies, University of Maryland,
    College Park, MD 20740, USA; marulraj@umd.edu (M.A.); Christopher.Grassotti@noaa.gov (C.G.);
    veljko@umd.edu (V.P.)
3   NOAA-NESDIS Center for Satellite Applications and Research, College Park, MD 20740, USA;
    Ralph.R.Ferraro@noaa.gov (R.F.); Bob.Kuligowski@noaa.gov (B.K.)
4   Cooperative Institute for Research in the Atmosphere, Colorado State University, Fort Collins, CO 80523, USA;
    shu-yan.liu@noaa.gov
5   NOAA/NCEP/Climate Prediction Center, College Park, MD 20740, USA; shaorong.wu@noaa.gov (S.W.);
    pingping.xie@noaa.gov (P.X.)
*   Correspondence: dmiller@unca.edu; Tel.: +1-828-232-5158

**Abstract:** Two heavy rainfall events occurring in early 2020 brought flooding, flash flooding, strong winds, and tornadoes to the southern Appalachian Mountains. Part I of the study examined large-scale atmospheric contributions to the atmospheric river-influenced events and subsequent societal impacts. Contrary to expectations based on previous work in this region, the event having a lower event accumulation and shorter duration resulted in a greater number of triggered landslides and prolonged downstream flooding outside of the mountains. One purpose of this study (Part II) is to examine the local atmospheric conditions contributing to the rather unusual surface response to the shorter duration heavy rainfall event of 12–13 April 2020. A second purpose of this study is to investigate the utility of several spaced-based QPE and vertical atmospheric profile methods in illuminating some of the atmospheric conditions unique to the April event. The embedded mesoscale convective elements in the warm sector of the April event were larger and of longer duration than of the other event in February 2020, leading to sustained periods of convective rain rates. The environment of the April event was convectively unstable, and the resulting available potential energy was sustained by relatively dry airstreams at the 700 hPa level, continuously overriding the moist air stream at low levels attributed to an atmospheric river.

**Keywords:** embedded mesoscale precipitation; extreme rainfall; landslides; southern Appalachian Mountains

## 1. Background

Part I of this study focused on the large-scale weather and surface features contributing to two heavy rainfall events in early 2020 triggering numerous landslides in the mountains of western North Carolina [1]. Space-based soil moisture observations of the upper (shallow) layer suggested sufficient recovery time occurred before the onset of the second (12–13 April 2020) heavy rainfall event, implying that earlier, potential pre-conditioning rain events in late March and early April 2020 were irrelevant to the triggering of the landslides that followed. The return to dry pre-storm soil moisture conditions of the upper layer within a few days is not unusual [1,2] due to the effects of wind and soil temperature on evapotranspiration. However, the mechanics of landslide initiation are often determined by soil moisture characteristics at deeper layers in the soil [2]. In addition to the

variable atmospheric factors of precipitation and evapotranspiration and the "constant" geologic factors of slope and lithology, the variable factors of runoff, percolation and storage of moisture and their interaction within the deeper soil layers determine if a region is predisposed to landslides when a storm of sufficient precipitation intensity can serve as the trigger. The latter factors involve change at relatively long time scales relative to the atmospheric factors due to the limited movement of water through porous soil. The concept of watershed "memory" in which the latter factors were linked to precipitation events in the recent and distant past was studied in the Coweeta River sub-Basin (CRB) of the southern Appalachian Mountains by Nippgen et al. [3]. A key finding was the significant lag correlation between monthly mean precipitation and monthly mean runoff ratio (runoff divided by precipitation) for monthly mean precipitation occurring up to six months before the monthly mean runoff ratio (cf., Figure 6 of [3]). Miller et al. [4] found a significant correlation between extreme (top 2.5%) precipitation events and landslide days occurring within 30 days of the events in the CRB. Hence, the implication is the long time scale of runoff in the CRB watershed contributes to a significant storage of water in the deeper soil layers for a prolonged period of time as percolation in this region is negligible due to the presence of impermeable bedrock located underneath the soil [3].

Of the two heavy rainfall events, the greatest number of landslides were triggered during or shortly after the April event, which had the lower total accumulation and shorter duration. In contrast, the 5–7 February 2020 event had a higher total accumulation and longer duration, even though a lesser number of landslides were triggered in the southern Appalachians during or just after the event. Post-case distant downstream flooding between Newport and Chattanooga, Tennessee (Figure 1) noted in Part I, with more significant (lesser) impacts observed after the April (February) 2020 event, suggested deep soil moisture storage was near or at (far from) capacity prior to the onset of heavy rainfall. The purpose of this second portion of the two-part study is twofold. First, to examine evidence that noteworthy atmospheric processes were responsible for triggering landslides initiated during or soon after passage of the heavy rainfall events in the southern Appalachians in early 2020. Second, to examine the utility of satellite QPEs and estimates of other atmospheric fields in highlighting noteworthy regional aspects of the events.

Although substantial progress has been made in understanding the atmospheric and geologic factors contributing to the initiation of landslides, reliable predictions of landslide initiation are still currently unattainable (e.g., [2,5]) due partly to a lack of understanding of the overall science and, to a greater degree, the lack of relevant earth and atmospheric observations covering remote areas in the mountains of the mid-latitudes. Recently, space-based observations of rain rate and surface properties (e.g., soil moisture) have reached horizontal resolutions useful to landslide scientists [2]. Comparisons of space-based soil moisture estimates by Thomas et al. [2] to in situ soil moisture observations in a landslide-prone region of California found that estimates from space were prone to soil moisture overestimates between major rain events. Space-based rainfall estimates are known to have their own unique challenges, particularly for mountainous regions (e.g., [6,7]). However, improvements in both types of space-based estimates have reached a point that an operational landslide risk assessment product has been developed with global applications [8].

The focus of this study is on the atmospheric side of the earth–atmosphere landslide initiation process of the two storms in early 2020. Numerous studies (e.g., [9]) have investigated a variety of weather events in the mid-latitudes providing ample hydrological input to overwhelm the soil's water redistribution and storage processes, reducing the shear resistance force of the soil below a critical threshold and resulting in slope failure [10]. Common methods used to quantify the critical threshold of the atmospheric rainfall contribution toward landslide initiation are the total accumulation within a 24-h period (e.g., [2,10]) and period rain rates exceeding a critical rain rate threshold (e.g., [10–12]). No matter the atmospheric critical threshold initiation methodology, landslides are almost always initiated in the southern Appalachian Mountains when taking a "direct hit" from rainfall of the spiral rainbands in the remnants of tropical cyclones (e.g., [9]). The challenge in

landslide initiation prediction, from the atmosphere side, is gaining a better understanding of initiation by the more common extratropical cyclones that occur during the cool season. As highlighted in Collins et al. [13], attention of weather events has often focused on the larger-scale aspects of the cool season storm, without attention to the individual effects of all precipitating systems embedded within the large storm system. Recently, investigators have focused on a "cause" and "trigger" period of atmospheric precipitation during which the former period conditions the soil and can either be distinct from or a part of the same storm providing the trigger period of precipitation [2,12].

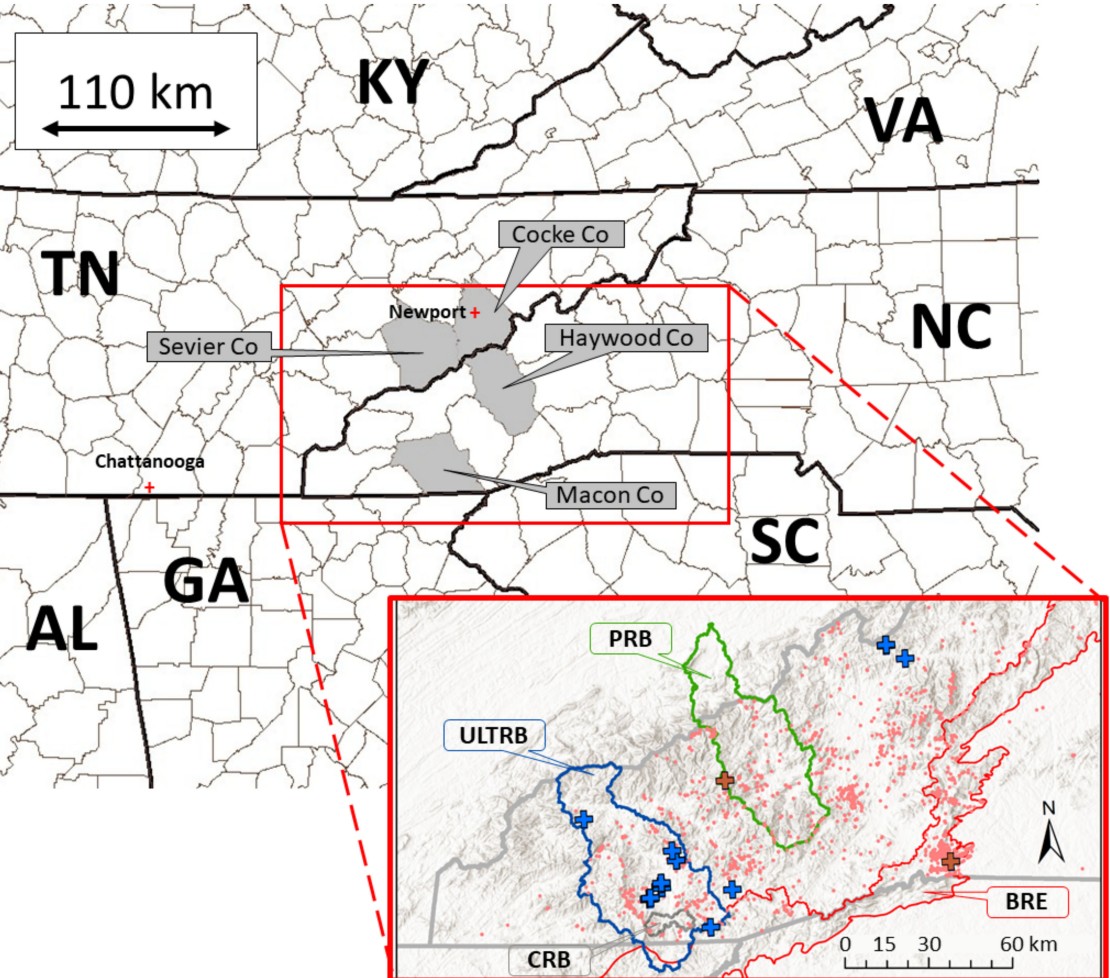

**Figure 1.** Locations of the Pigeon River Basin (PRB, green outline) and Coweeta River sub-Basin (CRB, gray outline), a sub-basin of the Upper Little Tennessee River Basin (ULTRB, blue outline), and topography (shaded) of the southern Appalachian Mountains. The Pigeon River Basin (PRB) corresponds to the borders of Haywood County, North Carolina and extends northward slightly into Cocke and Sevier Counties, Tennessee. The Coweeta River Basin is located in Macon County, North Carolina. Specifics on the locations and elevations of individual rain gauges of the Duke GSMRGN, located in the North Carolina region of the PRB, and CHLRGN, located in the CRB, are provided in Table A1. The center points of the PRB and CRB are located 60 km apart. The Blue Ridge Escarpment (labeled "BRE" and outlined in red) is the boundary between the Blue Ridge and the Piedmont physiographic province. The brown (blue) color-filled "+" symbols highlight two (21) landslide locations documented by the NCGS initiated by the 5–7 February (12–13 April) 2020 heavy rainfall event. Coral dots highlight landslide locations initiated since 1940 not occurring in February or April 2020. Locations of Newport and Chattanooga, Tennessee are also highlighted with a red "+" symbol.

## 2. Methodology

Observations from two rain gauge networks, archived atmospheric analyses of the Global Forecast System (GFS), space-borne estimates of quantitative precipitation (QPEs)

and vertical profiles of temperature, water vapor, and equivalent potential temperature, and landslides in the Southern Appalachian Mountains documented by the North Carolina Geological Survey (NCGS) serve as the primary datasets upon which are formulated the regional conclusions in Part II of this study in the southeastern U.S.

### 2.1. Surface-Based Observations

Rainfall observations of two rain gauge networks located in the Pigeon River Basin (PRB, Figure 1, Table A1) and the Coweeta River sub-Basin (CRB, Figure 1, Table A1) of the southern Appalachian Mountains, known hereafter as the Duke Great Smoky Mountains Rain Gauge Network (Duke GSMRGN) and the Coweeta Hydrologic Laboratory Rain Gauge Network (CHLRGN), serve as the reference datasets for defining event severity in early 2020. Observations from the Duke GSMRGN and CHLRGN used in this study have been collected for 11 and 86 years, respectively, and were also utilized in Part I of the project analysis. A listing of abbreviations unique to the study are included in Table A2 of Appendix B.

Observations of the Duke GSMRGN are used primarily to illuminate the variability of the two impactful events investigated in this study as the 32 rain gauges are located at elevations varying from 1036 to 2003 m, covering the PRB area (1823 km$^2$). The closer proximity of the CHLRGN to the Blue Ridge Escarpment (Figure 1) allows investigation of a potential enhancement of rainfall observed during the two events under favorable wind conditions. Its nine rain gauges cover a smaller elevation range (687 to 1366 m) and area (16.3 km$^2$) compared to the Duke GSMRGN.

Following the methodology of Part I, total rainfall accumulation observed by the two rain gauge networks was binned into synoptic 6-h periods (0000, 0600, 1200, and 1800 coordinated universal time (UTC)) corresponding to the 6-h time resolution of the GFS historical analysis of the National Centers for Environmental Information archives. Events were defined as having concluded when no amounts were recorded at any of the network gauges during at least a single synoptic 6-h period [14]. Non-zero per gauge accumulation amounts of each consecutive synoptic 6-h period were added to calculate the event total per gauge accumulation. Events at each gauge network were defined separately to capture the influence on precipitation production of local orography.

Landslide inventory data for North Carolina used in the study came from the landslide geodatabase maintained by the NCGS [15]. The geodatabase documents 23 landslides of various types for the February–April 2020 focus period of this study (color-filled "+" symbols in Figure 1), where the known date(s) of movement for individual landslides are recorded in the geodatabase.

### 2.2. Event Rainfall–Landslide RRt "Profile" Algorithm

The hourly rain rate and time (RRt) "profile" template of Figure 2a consists of two periods; an earlier lighter precipitation phase ($\Delta t1$) and a later heavier precipitation phase ($\Delta t2$). Hereafter, we will borrow the terminology of Bogaard and Greco [12] and Thomas et al. [2] and refer to the $\Delta t1$ and $\Delta t2$ periods of the RRt profile as the "cause" and "trigger" phases, respectively. Context of when these phases occur in a passing extratropical cyclone are indicated in Figure 2b, although its original application by Nagle and Serebreny [16] and Medina et al. [17] was for oceanic cyclones making landfall in the eastern Pacific Ocean. The general concepts still apply for continental extratropical cyclones, whereby the cause phase occurs during the early and first-half middle storm sectors, and the trigger occurs during the second-half of the middle storm sector (black shaded banding in Figure 2b). The mean hourly rain rate during the cause phase ("RR1" in Figure 2a) represents rainfall consistent with an upglide of warm humid air along the warm front of a surface cyclone, in the vicinity of strong warm air advection. After passage of the surface warm front, the conclusion of the early storm sector, warm air advection weakens within the storm warm sector, but precipitation intensity gradually increases during the first half of the middle storm sector. Rainfall during the cause phase gradually moves the watershed deep soil

moisture storage closer to its capacity, offsetting runoff and evapotranspiration effects over the period leading up to the event. The end of the cause phase marks a transition to a heavier hourly mean rain rate ("ΔRR" in Figure 2a), noted by black banded features in Figure 2b, that correspond to the second-half of the middle storm sector, the trigger phase. Hourly rain rates drop-off significantly after the trigger phase, when the late storm sector moves into the region, typically after the surface cold front has passed and cold air advection is present.

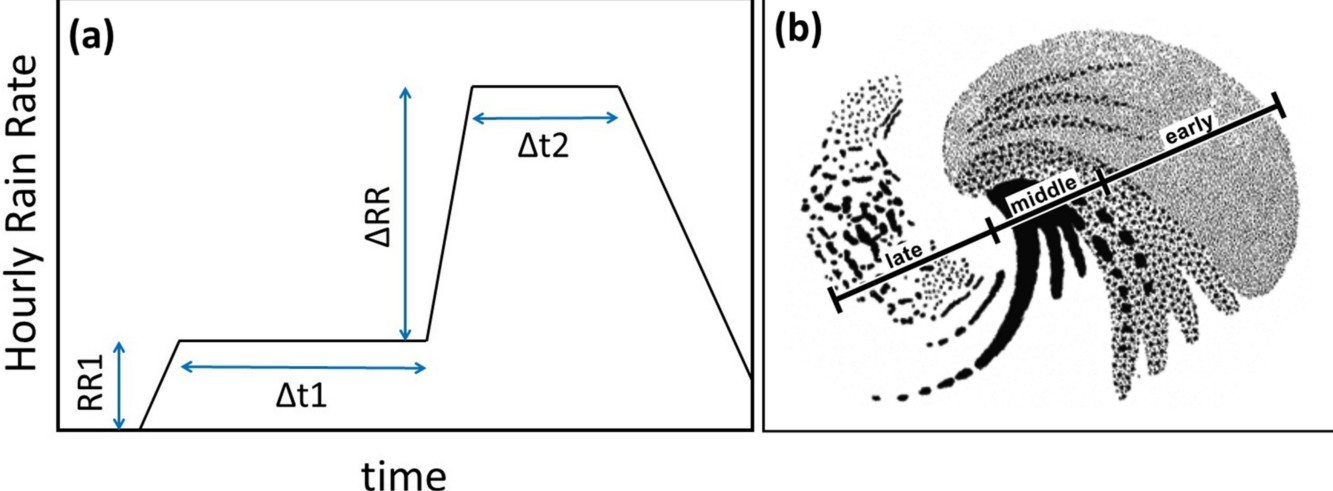

**Figure 2.** Conceptual model showing (**a**) hourly rain rate and time (RRt) profile during (**b**) a single extratropical synoptic scale cyclone event consisting of the "cause" (Δt1) and "trigger" (Δt2) phases of a nearby landslide. The landslide would initiate either during the "trigger" phase or shortly thereafter. Panel (**b**) is Figure 1 of Medina et al. [17] (© American Meteorological Society. Used with permission.), adapted from Nagle and Serebreny [16], where the precipitation intensity is indicated by the degree of shading. Line segments indicate the early, middle, and late sectors of the storm.

If a landslide is to occur, the soil has been conditioned before and/or during the cause phase to the point where its means of disbursing accumulating rain is nearly overwhelmed (close to saturation). Once the trigger period starts, the critical hydrological pressure threshold for a particular location is exceeded and the landslide is initiated either immediately or after a brief period. The heavier rain rates of the trigger period are typically associated with occasional mesoscale convective elements (MCEs) corresponding to the pre-cold frontal warm sector and narrow cold frontal rainbands ahead of the cold front present in an extratropical cyclone [13].

The four parameters of the RRt profile (RR1, Δt1, delta ΔRR, and Δt2) of Figure 2a are highly variable in space and time. The critical cause threshold is RR1 × Δt1 and the critical trigger threshold is ΔRR × Δt2. The critical cause threshold is a function of the local surface characteristics (e.g., antecedent mid- and lower-soil (deep) layer moisture, soil type, lithology, slope), while the critical trigger threshold is a function of event rainfall during the cause phase and antecedent soil moisture. Most extratropical synoptic scale storms move through the southern Appalachians so rapidly that the critical thresholds are not met due to the brief cause (Δt1) and/or trigger (Δt2) periods and/or low-intensity rain rates (RR1, ΔRR).

Rainfall observations in the PRB (Duke GSMRGN) and the CRB (CHLRGN) were analyzed to create RRt profiles at each rain gauge during the two heavy rainfall events in early 2020. The profiles were constructed "backwards" in the sense that the trigger phase hourly mean rain rate was determined first by maximizing the summed hourly accumulation divided by the number of hours contributing to the summed hourly accumulation. The maximizing process worked backwards in time from rainfall observations at the conclusion of each event. Once the time of the trigger phase initiation was determined, a similar backwards maximizing process was used to determine the start of the cause phase. Once

maximized, the cause phase hourly mean rain rate was simply the hourly accumulation during the cause phase divided by the number of cause phase hours. The conclusion of the trigger phase was the hour when the hourly accumulation toward the end of the event fell below the cause phase hourly mean rain rate. Hourly accumulation was used to generate the RRt profiles as this was the finest time resolution available from the CHLRGN observation archive.

### 2.3. In Situ Rainfall Observations and Space-Based QPE Comparisons

To avoid the complications of point-versus-area precipitation comparisons between Duke GSMRGN gauge observations and satellite QPEs, a $1° \times 1°$ landslide focus region (34.75°N to 35.75°N, 83.5°W to 82.5°W, Figure 4d; where most of the landslides in February and April 2020 occurred, Figure 1) was defined for calculating area-averaged accumulation and rain rate, maximum rain rate, and rain rate standard deviation for pixels or gauges located within the focus region. CHLRGN gauge observations were excluded from the comparison as the finest time resolution of the archived dataset was one hour. For the time of each satellite QPE, Duke GSMRGN 15 min rain rate observations were averaged over an advective time scale. The time scale was computed by converting the GFS-analyzed 700 hPa level wind speed and direction at a grid point in the middle of the southern landslide focus region boundary to the transit time for an air parcel entering the southern boundary and exiting the northern boundary. A linear interpolation of GFS 700 hPa level wind speed and direction at 6-h synoptic times was used to determine a 700 hPa wind vector at asynoptic times typical of satellite observations. An adjustment was made for the shift in the 700 hPa level wind direction from south and variability in the east–west and north–south distance for one degree latitude and longitude at the center point of the focus region (35.25°N, 83.0°W). The advective time scale was split such that Duke GSMRGN observations were averaged between half the advective time scale before and half after the time of each satellite QPE snapshot over the landslide focus region.

### 2.4. Space-Borne Observations

#### 2.4.1. Microwave Integrated Retrieval System (MiRS)

The microwave integrated retrieval system, a passive microwave retrieval algorithm, has been run operationally at NOAA since 2007. Compared to visible and infrared radiation, microwaves have a longer wavelength and, thus, can penetrate through the atmosphere more effectively. This feature allows microwave observations under almost all weather conditions including in cloudy and rainy atmospheres. MiRS follows a 1-dimensional variational (1DVAR) methodology [18,19]. The inversion is an iterative physical algorithm in which the fundamental physical attributes affecting the microwave observations are retrieved physically, including the profiles of atmospheric temperature, water vapor, non-precipitating cloud, hydrometeors, as well as surface emissivity and skin temperature [20]. The Joint Center for Satellite Data Assimilation (JCSDA) Community Radiative Transfer Model (CRTM) [21,22] was used as the forward and Jacobian (i.e., radiance derivatives with respect to the geophysical parameters) operator to simulate the radiances at each iteration prior to fitting the measurements to within the combined instrument and forward model noise level. After the core parameters of the state vector are retrieved in the 1DVAR step, additional post-processing was performed to retrieve derived parameters based on inputs from the core 1DVAR retrieval. The MiRS precipitation rate was determined as a post-processing step that relates profiles of the core retrieved hydrometeors (i.e., rain and ice water) to the surface precipitation. MiRS precipitation rates from a number of satellites have been extensively validated using ground-based references [23–25].

MiRS retrievals from both Suomi-NPP and NOAA-20 ATMS measurements were used in this study [26,27]. Vertical cross-sections of water vapor were constructed based on the MiRS-retrieved water vapor mixing ratios at each ATMS field of view (FOV). The retrieved profiles were specified on 100 pressure layers from 1085 to 0.01 hPa. For locations covered within the ATMS measurement swath of 2600 km, there was a vertical sounding available

every 15 km, which is the horizontal spacing between ATMS FOVs. At the latitude of the study region, there were normally two overpasses per day per satellite.

The cross-sections of equivalent potential temperature ($\theta$e) were derived directly from the MiRS-retrieved profiles of water vapor and temperature and the corresponding pressure values at each vertical layer. The formulation of Bolton [28] was used (i.e., Equation (43)) to compute $\theta$e.

### 2.4.2. Climate Prediction Center Morphing Technique (CMORPH), Second Generation

The second-generation Climate Prediction Center (CPC) morphing technique (CMORPH, [29]) was constructed through integrating information from passive microwave (PMW) retrievals from low earth orbit (LEO) satellites, infrared (IR) observed cloud top temperature from geostationary platforms, as well as gauge-based analysis of daily precipitation [30]. First, PMW precipitation retrievals from all available LEO sensors were calibrated against those from a reference sensor, the global precipitation measurement (GPM) mission microwave imager (GMI). These inter-calibrated PMW retrievals were composited to form combined global fields of PMW retrievals of a 30-min precipitation, called MWCOMB. Estimates of 30-min precipitation were derived from the GEO IR data through calibration against the MWCOMB and utilized to compute the cloud motion vectors. PMW retrievals of 30-min precipitation rates as documented in MWCOMB were propagated from their respective measurement time to the target analysis along the cloud motion vectors to form a global field of 30-min precipitation, producing the purely satellite-based raw CMORPH satellite precipitation estimates. Bias in the raw CMORPH was removed through calibration against the CPC daily gauge analysis over land and against the Global Precipitation Climatology Project merged analysis V3.1 over the ocean.

### 2.4.3. Goddard Profiling Algorithm (GPROF)

The most recent version of Goddard PROFiling (GPROFv2017) algorithm [31–34] run at NOAA/NESDIS was employed to retrieve precipitation rate from GCOMW1-AMSR2 brightness temperature (Tb) observations. Using a Bayesian approach [35], the profiling algorithm inverted the level-1b AMSR2 product to estimate the probability of precipitation rate relying on an a priori knowledge of the relationship between the observed Tb and atmospheric state. Details on this inversion process are provided in Kummerow et al. [33].

The a priori knowledge was exposed to the retrieval via a database built from globally observed precipitation rates coupled with corresponding Tbs. Comparing the observed Tb vector against this global database allowed the inversion of radiances to precipitation rate. In a relatively simple process of comparing the distance in Tb-space between the observed and a priori stored Tb vectors, each database element received a weight to reflect its proximity (i.e., similarity) to the observed Tb vector. These weights were then used to average corresponding a priori precipitation values (i.e., rates or atmospheric states) and formed the retrieval's best estimate of surface rainfall.

The a priori knowledge of the relationship between the observed radiances and surface precipitation rates was built from the GPM dual frequency precipitation radar (DPR) Ku observed hydrometeor profiles and corresponding simulated Tb. To ensure robust simulations, the radiative transfer model relied on ancillary information such as surface type, modeled atmospheric total precipitable water (TPW) and 2-m temperature (T2m). The surface types were derived from an emissivity climatology [36], while TPW and T2m parameters were obtained from widely available reanalysis datasets such as ERA5 and GFS. The Tb used in this study comes from GCOM-AMSR2 level-1B dataset routinely generated at NOAA/NESDIS. Using a total of ten of GCOM-AMSR2 channels (10–89 GHz, dual pol), GPROF retrieval provided instantaneous precipitation estimates at approximately 7.5 km spatial resolution with a 10 km field of view. Identification of precipitation phase follows that of Sims and Liu [37], while the precipitation type was assigned based on the dominant typology among the weighted database elements.

### 2.4.4. Enterprise

The GOES-R Enterprise rainfall rate algorithm (also known as self-calibrating multivariate precipitation retrieval, or SCaMPR) was based on IR data from geostationary satellites that were calibrated against PMW rain rate retrievals. This strategy allowed the calibration to vary in space and time as needed while keeping data latency to a minimum, which is especially critical for flash flood applications. For additional details, please refer to Kuligowski et al. [38].

## 3. Results

The 6-h synoptic periods covering observed rainfall by the Duke GSMRGN during the two events in early 2020 spanned the periods 0600 UTC 5 February–1200 UTC 7 February (54 h) and 1200 UTC 12 April–1800 UTC 13 April 2020 (30 h). A description of the synoptic scale weather pattern during each event is provided in Part I of the study. The focus of this part is on a regional view in the southeastern U.S. of the weather pattern at times corresponding to the cause and trigger phases of each event. Selected periods during the trigger phases of each event corresponded to the time when the associated atmospheric river (AR) of moisture was centered on the southern Appalachian Mountains; 1200 UTC 6 February 2020 and 0600 UTC 13 April 2020 (cf., Figure 6a,b of Part I). Selected cause periods of each event corresponded to the nearest 6-hourly time period of the GFS gridded analyses when widespread precipitation was observed in the PRB early in the event; 1800 UTC 5 tFebruary 2020 and 1800 UTC 12 April 2020 (Figure 3).

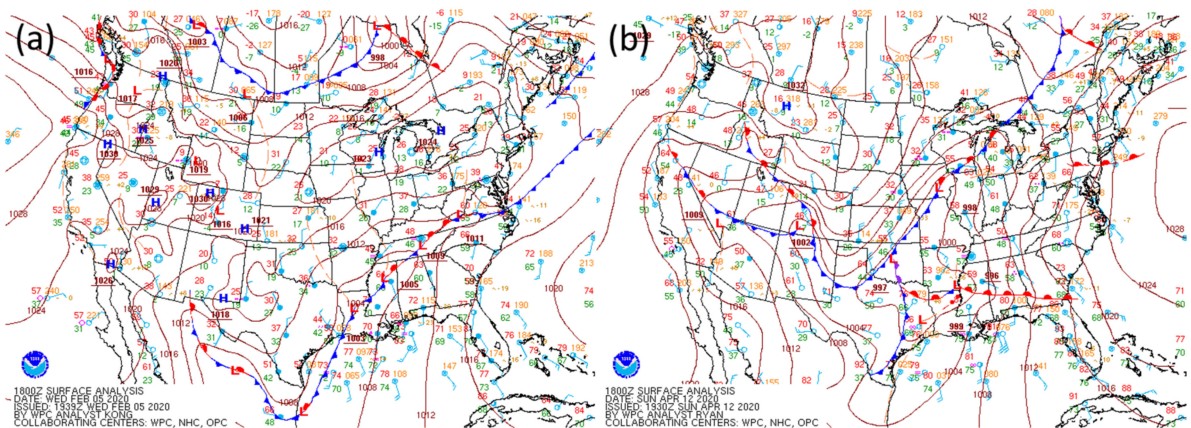

**Figure 3.** Sea level pressure maps and frontal analysis of the Weather Prediction Center at (**a**) 1800 UTC 5 February 2020 and (**b**) 1800 UTC 12 April 2020 (accessed online at https://www.wpc.ncep.noaa.gov/archives/web_pages/sfc/sfc_archive.php on 10 March 2021).

The cause phase of the February 2020 event differed from the schematic of Figure 2b in that the southern Appalachian Mountains were entirely on the warm-air side of the frontal zone located in eastern Tennessee and northern Alabama (Figure 3a) during the early sector of the storm. Hence, its cause period corresponded more directly to the first-half middle sector of the storm schematic in which weak warm air advection occurred over the southern Appalachians and, from the synoptic scale perspective, contributed to broad ascent of low-level humid air over the interior southeastern U.S. (Figure 4a), resulting in sporadic precipitation (Figure 4c). This period represented a transition between a recently departed AR of 4 February and a second AR corresponding to the heavy rainfall of 6 February. Hence, the relatively low 700 hPa level equivalent potential temperature ($\theta$e) values offshore of the southeastern United States (Figure 4b) were a reflection of the transitory anticyclone before the 6 February storm had moved into the region, the leading edge of which was evident over Alabama, southcentral Tennessee, and Georgia where high 700 hPa level $\theta$e values were collocated with winds exceeding 20 m s$^{-1}$.

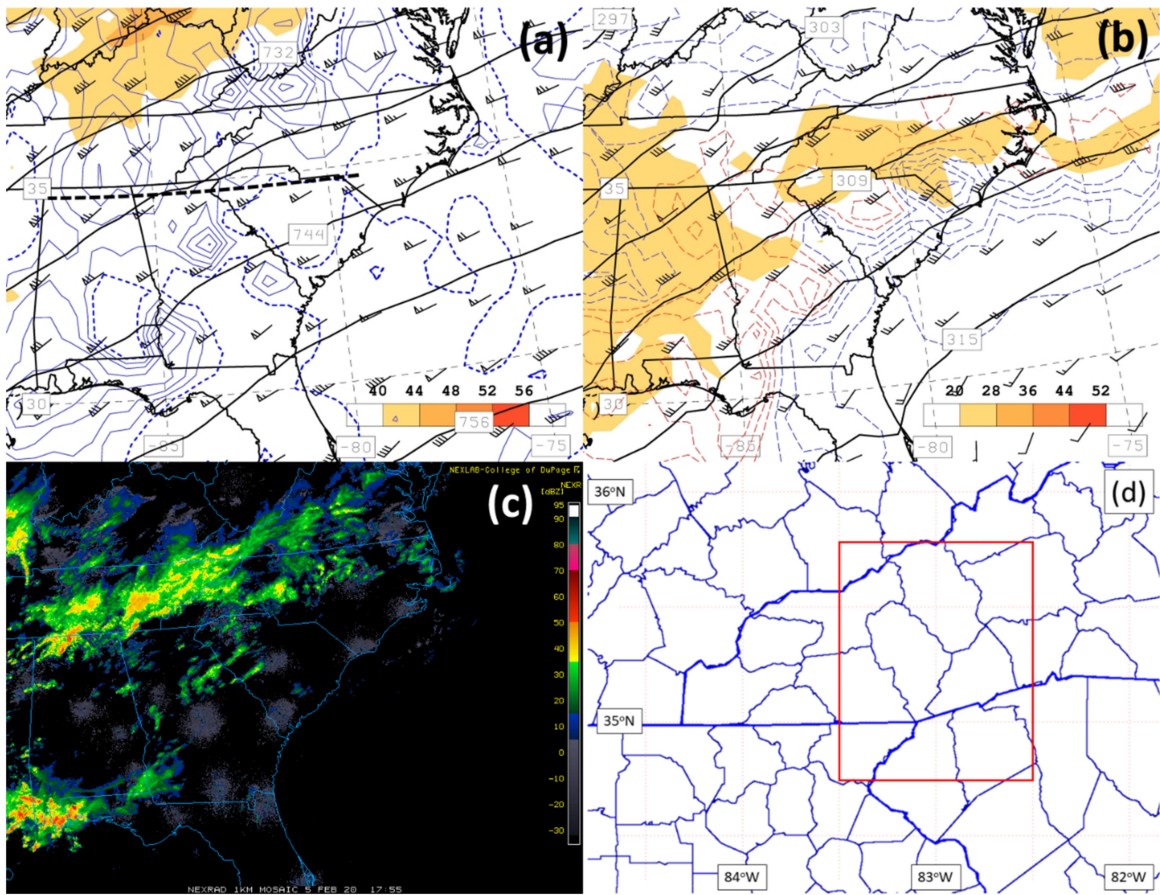

**Figure 4.** GFS-analyzed fields valid at 1800 UTC 5 February 2020 of: (**a**) 400 hPa level geopotential height (dam, solid black contours), wind speed (m s$^{-1}$, shading), and wind vectors (kt) and 500 hPa level rising motion × 10$^{-3}$ (hPa s$^{-1}$, blue contours; thick dashed contour is zero vertical motion) and (**b**) 700 hPa level geopotential height (dam, solid black contours), wind speed (m s$^{-1}$, shading), equivalent potential temperature (K, final blue (first red) dashed contour value is 321 K (324 K)), and wind vectors (kt). WSR88D (**c**) composite reflectivity (dBZ) courtesy of the College of DuPage (accessed online at https://www2.mmm.ucar.edu/imagearchive/ on 10 March 2021). Thick dashed line in panel (a) marks the position of the vertical cross sections displayed in Figures 8 and 9 oriented along 34.75°N, extending from 88 to 79°W. The red outline of panel (**d**) represents the boundary of the 1° × 1° landslide focus region (34.75°N to 35.75°N, 83.5°W to 82.5°W) utilized in making area-averaged rainfall comparisons.

The cause phase of the April 2020 event (Figure 3b) more closely resembled the early sector of the Figure 2b schematic in which strong warm air advection and overrunning along the warm front (Figure 5a,b) contributed to sporadic moderate precipitation in the southern Appalachians (Figure 5c). The AR associated with the heavy rainfall of 13 April had already entered the southern Appalachians (Figure 5b) during the cause phase, evident by the region of high-$\theta$e values collocated with 700 hPa level wind speeds ranging between 20 and 44 m s$^{-1}$ to the southwest. Additionally evident at this time was the "belt" of high-$\theta$e air (Figure 5b) identified in the ALPW imagery of Part I emanating outward from the strong surface anticyclone offshore of Florida and extending inland over the Georgia–South Carolina border (cf., Figure 11b,d of Part I). Convergence of humid air streams at low levels has been identified as a signature in other heavy rainfall events [39].

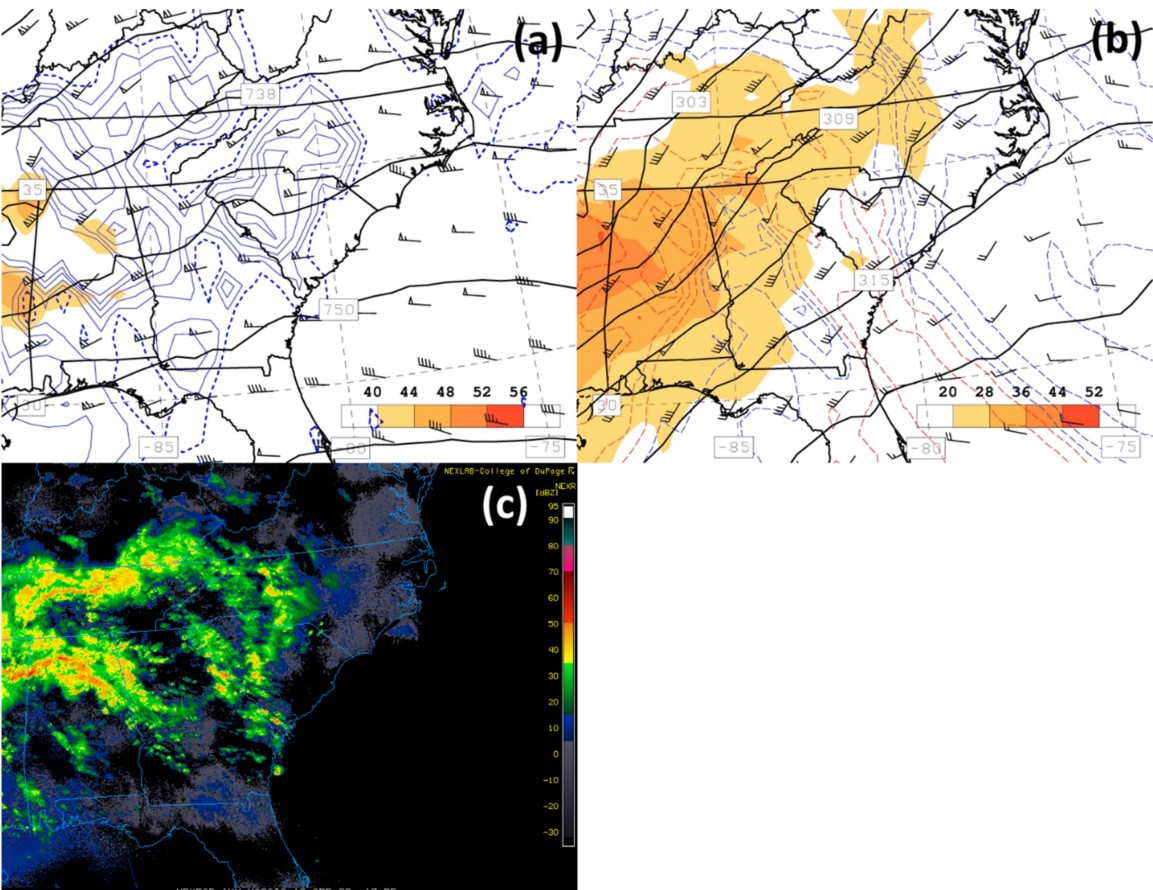

**Figure 5.** As in Figure 4, except valid at 1800 UTC 12 April 2020 of: (**a**) 400 hPa level geopotential height (dam, solid black contours), wind speed (m s$^{-1}$, shading), and wind vectors (kt) and 500 hPa level rising motion $\times$ 10$^{-3}$ (hPa s$^{-1}$, blue contours; thick dashed contour is zero vertical motion) and (**b**) 700 hPa level geopotential height (dam, solid black contours), wind speed (m s$^{-1}$, shading), equivalent potential temperature (K, final blue (first red) dashed contour value is 321 K (324 K)), and wind vectors (kt). WSR88D (**c**) composite reflectivity (dBZ) courtesy of the College of DuPage (accessed online at https://www2.mmm.ucar.edu/imagearchive/ on 10 March 2021).

Consistent with the schematic of Figure 2b, the trigger phase of the February 2020 event resembled the increasingly convective and heavy rainfall of the second-half middle storm sector (Figure 6c) compared to earlier in the event. By this time, the AR center was aligned with the southern Appalachians, highlighted at the 700 hPa level as a ridge of high-$\theta$e values collocated within a wide swath of strong winds between 20 and 40 m s$^{-1}$ (Figure 6b). The seemingly random convective nature of the storm was evident in variations of wind speed located to the southeast of the merged polar/sub-tropical jet core, as numerous pockets of sub-20 m s$^{-1}$ speeds were found at the 400 hPa level (Figure 6a) above maxima of ascending motion associated with embedded MCEs (Figure 6a,c). As the second-half middle storm sector passed over the southern Appalachians, convection remained rather disorganized and lacking in intensity until its leading edge moved past the mountains at 1425 UTC 6 February and a convective line developed along an elongated outflow boundary by 1855 UTC 6 February extending from central North Carolina to the panhandle of Florida (not shown).

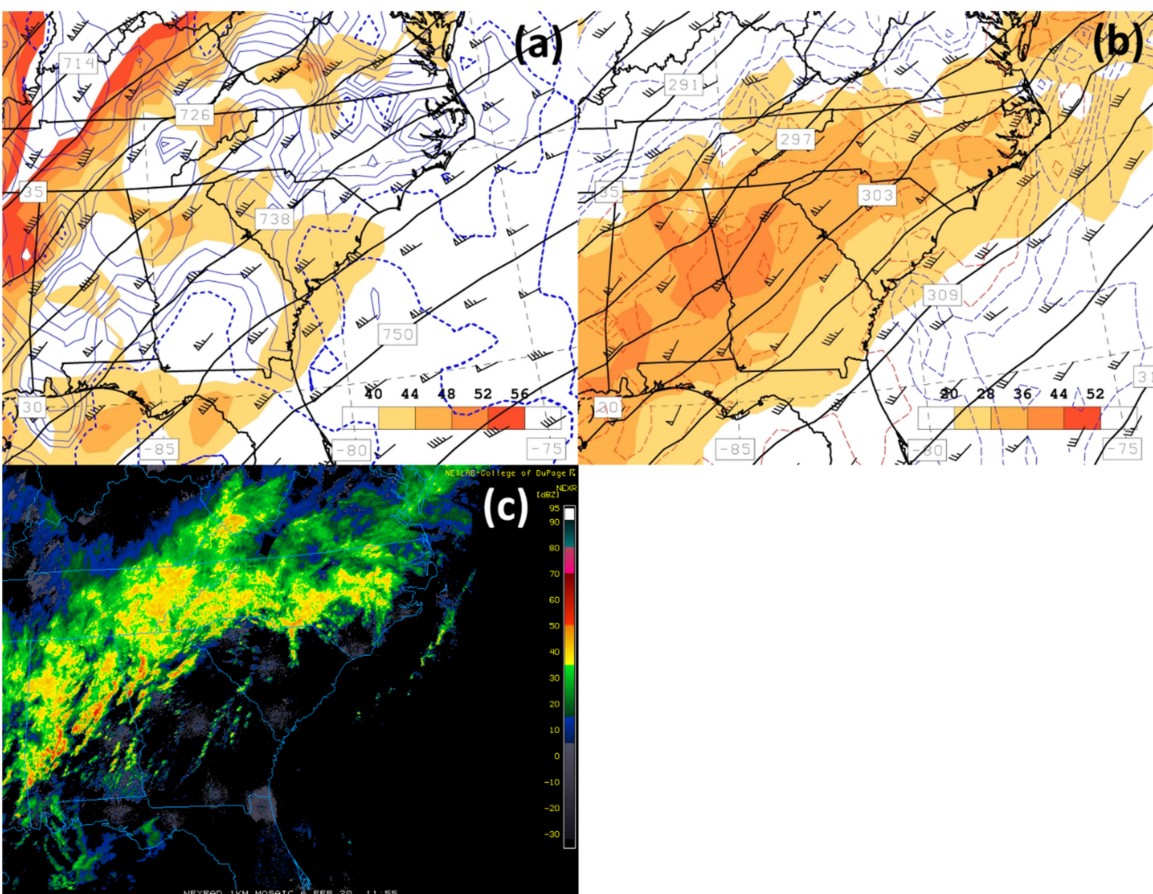

**Figure 6.** As in Figure 4, except valid at 1200 UTC 6 February 2020 of: (**a**) 400 hPa level geopotential height (dam, solid black contours), wind speed (m s$^{-1}$, shading), and wind vectors (kt) and 500 hPa level rising motion x 10$^{-3}$ (hPa s$^{-1}$, blue contours; thick dashed contour is zero vertical motion) and (**b**) 700 hPa level geopotential height (dam, solid black contours), wind speed (m s$^{-1}$, shading), equivalent potential temperature (K, final blue (first red) dashed contour value is 321 K (324 K)), and wind vectors (kt). WSR88D (**c**) composite reflectivity (dBZ) courtesy of the College of DuPage (accessed online at https://www2.mmm.ucar.edu/imagearchive/ on 10 March 2021).

Although the second-half middle storm sector during the trigger phase of the April 2020 event moved through the southern Appalachians rather quickly, the convection was organized into larger-scale elements over the mountains, particularly near the North Carolina, Georgia, and South Carolina border (Figure 7c). Strong winds at the 700 hPa level, associated with the AR, moved humid air northward along a narrow corridor (Figure 7b) providing fuel for the narrow swath of observed convection (Figure 7c). Noteworthy is the tongue of dry air whose axis was located over South Carolina and Georgia, on the anticyclonic shear side of the 700 hPa level jet core and a secondary tongue of dry air over northwestern Alabama. As will be shown, the source region of the dry air in the two tongues of low $\theta e$ was different, but each was significant in contributing directly and indirectly to the evolution of intense convection over the southern mountains. The "split" in the sub-tropical jet core, evident at the 400 hPa level (Figure 7a), was a consequence of the narrow zone of intense convection present over eastern Tennessee, northern Georgia, and southeastern Alabama (Figure 7c) whose strong updrafts (Figure 7a), divergence aloft, and sensible heating due to latent heat release disrupted the jet dynamics.

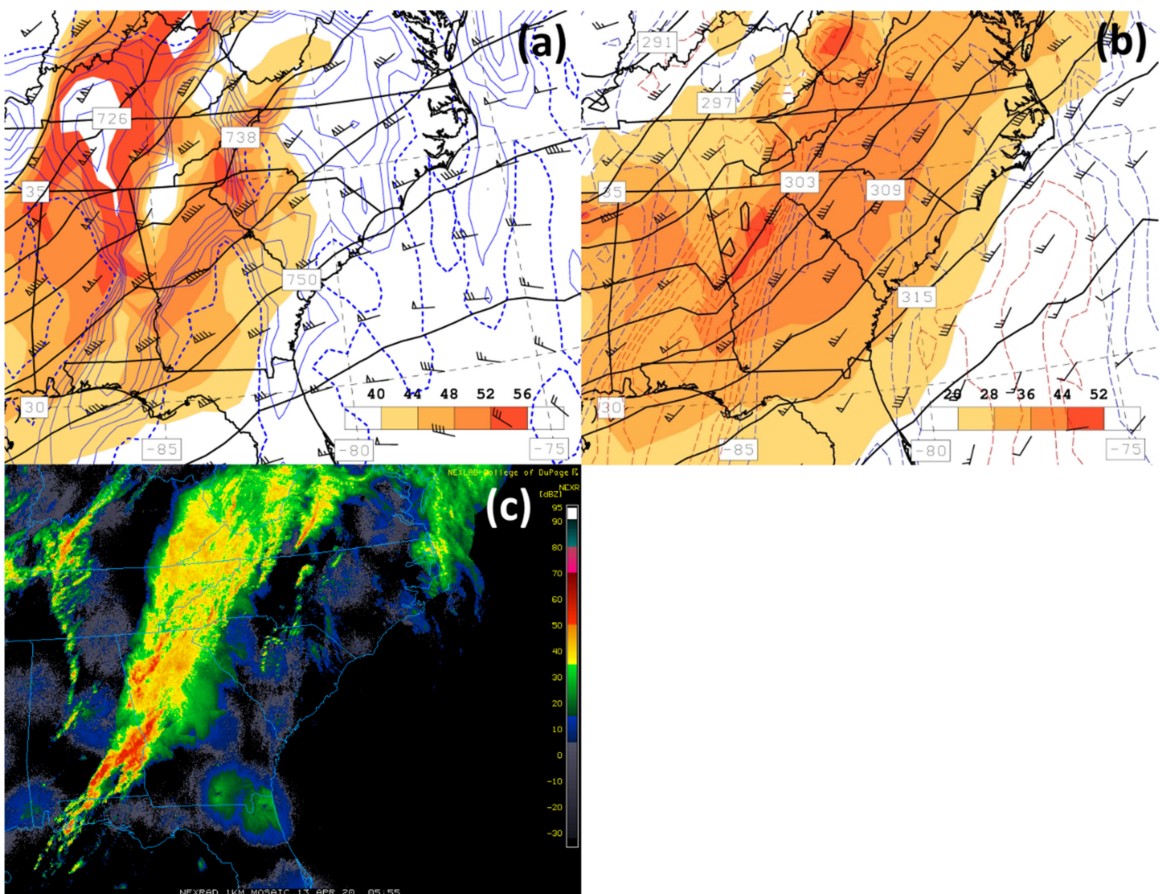

**Figure 7.** As in Figure 4, except valid at 0600 UTC 13 April 2020 of: (**a**) 400 hPa level geopotential height (dam, solid black contours), wind speed (m s$^{-1}$, shading), and wind vectors (kt) and 500 hPa level rising motion $\times$ 10$^{-3}$ (hPa s$^{-1}$, blue contours; thick dashed contour is zero vertical motion) and (**b**) 700 hPa level geopotential height (dam, solid black contours), wind speed (m s$^{-1}$, shading), equivalent potential temperature (K, final blue (first red) dashed contour value is 321 K (324 K)), and wind vectors (kt). WSR88D (**c**) composite reflectivity (dBZ) courtesy of the College of DuPage (accessed online at https://www2.mmm.ucar.edu/imagearchive/ on 10 March 2021).

Vertical cross sections extending from 34.75°N, 88°W (left) to 34.75°N, 79°W (right, section location is plotted in Figure 4a) both valid six hours before (0600 UTC 6 February, Figure 8a) and at the time of the AR being centered on the southern Appalachians (1200 UTC 6 February, Figure 8b), during the trigger phase of the February 2020 event, are displayed in Figure 8. The 6-hourly progression of the merged polar/ sub-tropical jet in the upper-left (western) corner of the sections reflected the relatively slow eastward propagation of the storm warm sector. The breadth of the stronger winds in the lower layer at 1200 UTC 6 February (Figure 8b) was a response to the strengthening baroclinic zone evident below the 700 hPa level and contributed to the enhanced integrated vapor transport within the AR (cf., Equation (1) of Part I). Small-scale structures evident as local minima in wind speed were collocated with small-scale convective elements highlighted by the ageostrophic circulation vectors showing strong rising motion. Isolated pockets of strong rising motion also reflected the weakly stratified environment of the warm sector at both times as highlighted by the wide vertical spacing between $\theta$e contours. Only the cold dome of air at the surface near the western boundary of the section, behind the surface cold front at 86°W (Figure 8b), had appreciable strong environmental stability.



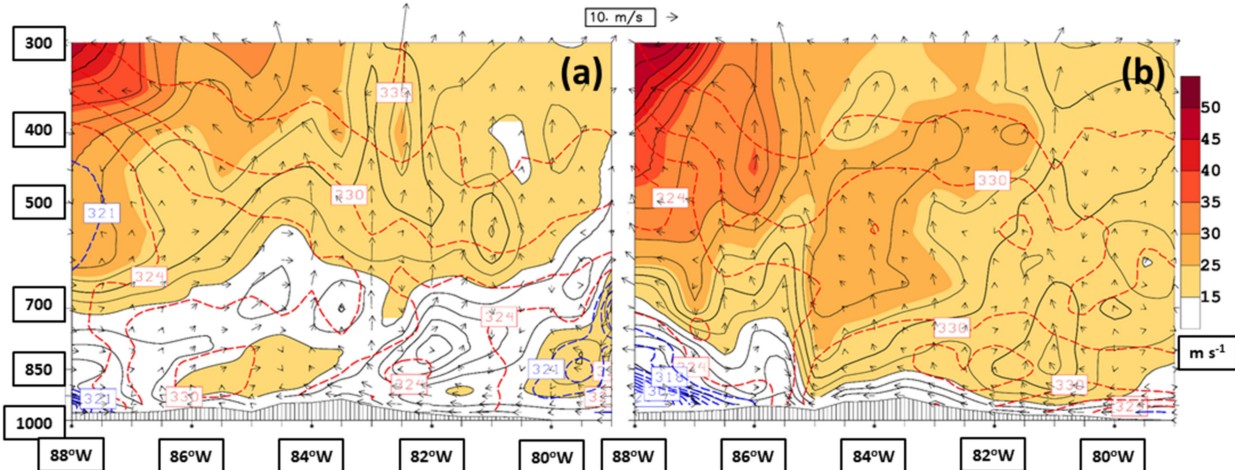

**Figure 8.** Vertical cross-section (endpoints at 34.75°N; 88°W (L), 79°W (R)) of GFS-analysed fields of wind speed normal to the section (m s$^{-1}$, shading and solid contours; thick contour is 21 m s$^{-1}$ isotach), equivalent potential temperature (K, final blue (first red) dashed contour value is 321 K (324 K)), and ageostrophic circulation in the section (arrow; reference horizontal ageostrophic wind of 10 m s$^{-1}$ is shown in middle top) valid at (**a**) 0600 UTC and (**b**) 1200 UTC 6 February 2020.

Comparable vertical cross sections of the April 2020 event valid six hours before (0000 UTC 13 April, Figure 9a) and at the time of the AR being centered on the southern Appalachians (0600 UTC 13 April, Figure 9b), both during the trigger phase of the event, show the fast progression of the storm warm sector as reflected by the passage of the sub-tropical jet core in the upper-left corner of the section (cf., Figure 9c of Part I). The jet cores at 84.5°W at 0000 UTC 13 April (Figure 9a) and at 83°W at 0600 UTC 13 April (Figure 9b) owed their existence in the upper (600–300 hPa) layer to the sub-tropical jet dynamics. The local minimum of wind speed in the upper layer immediately east of the sub-tropical jet core was a consequence of MCEs having strong rising motion, made possible by the weakly stratified environment of the warm sector.

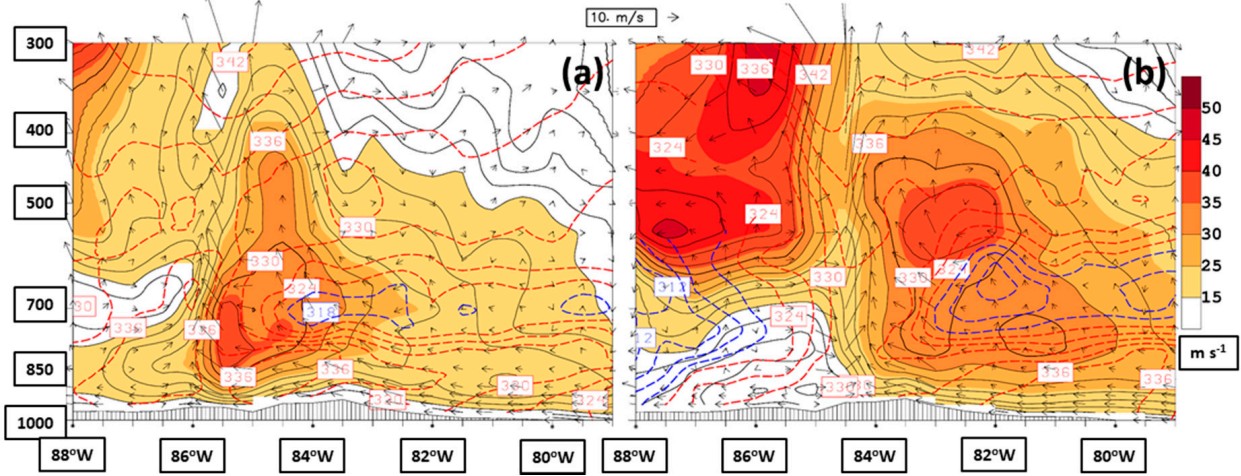

**Figure 9.** As in Figure 8, except valid at (**a**) 0000 UTC and (**b**) 0600 UTC 13 April 2020 and thick contour of wind speed normal to the section corresponds to the 33 m s$^{-1}$ isotach.

The jet core maxima in the lower (1000–600 hPa) layer of the April 2020 event represented a thermal wind response to a reversal of the east–west horizontal density gradient evident as one moves upward from the ground to the 600 hPa level. Near the ground, humid low-density (high-$\theta e$) air was moving into the eastern half of the section, forced by flow about the surface anticyclone to the southeast and identified in the ALPW imagery (cf., Figure 11b,d) of Part I and in Figure 5b. The east–west horizontal density gradient

reversed at ~800 hPa level as dry high-density air was advected into the section at the 700 hPa level, corresponding to the eastern "dry tongue" as described in the discussion of Figure 7b. Close inspection of Figure 9 reveals three distinct lower layer jet core maxima at 85.5°W at 0000 UTC 13 April (Figure 9a) and 83°W and 81.5°W at 0600 UTC 13 April (Figure 9b), highlighted by the thick 33 m s$^{-1}$ isotach. The cores at 85.5°W and 83°W were found on the leading edge of strong and elongated convective elements (Figure 7c), drifting eastward and weakening as their associated convection dissipated (e.g., lower layer jet core at 81.5°W in Figure 7b). Each low-level jet enhanced the observed rainfall during the trigger phase of the April 2020 event in two ways; by increasing the horizontal vapor transport into the southern Appalachians, making vapor available to ascending air forced by synoptic-scale processes (e.g., cyclonic vorticity and warm air advection; cf., Figure 9a of Part I, upper-level divergence in the right entrance quadrant of the sub-tropical jet streak; Figure 7a), and by providing water vapor fuel (latent heat) for driving the intense convection. A wide swath of convective instability of the environment in the storm warm sector, capped by the eastern dry tongues at 0000 and 0600 UTC 13 April, was relatively long-lived during the April 2020 event and created favorable conditions for heavy rainfall.

The western dry tongue at 0600 UTC 13 April, apparent at the western boundary of the section (Figure 9b), extended through a relatively deep layer, overrode high-$\theta$e air at the surface, and represented the approach of drier and cooler air from the west as part of the late storm sector. This zone of convective instability, positioned along the flanking edge of the storm warm sector, was long-lived as vertical differences in layered precipitable water were evident over western Mississippi at 0000 UTC 13 April 2020 in ALPW imagery (cf., Figures 11b,c and 12d) of Part I.

The source of high-density dry air noted at the ~700 hPa level from 0000–0600 UTC 13 April 2020 (Figure 9) was analyzed using the HYSPLIT trajectory model [40], its results displayed in Figure 10. Air parcel 72-h trajectories ending at the 700 and 850 hPa level for locations east of the sub-tropical jet core are displayed in Figure 10a (end point; 34.75°N, 84°W at 0000 UTC) and Figure 10b (end point; 34.75°W, 82°W at 0600 UTC) showed similar points of origin for the 700 hPa level air parcels ending in the eastern dry tongue (Figure 7b). Both air parcels started within the 700–800 hPa layer for locations just south of Cuba and moved anticyclonically with gradual subsidence associated with the offshore anticyclone until making landfall. The relative humidity of the 0000 UTC- and 1200 UTC- arriving air parcel, as established from GFS analyses, just after making landfall over Florida was 30 and 10%, respectively. Inspection of soundings released at the Owen Roberts Airport (19.30°N, 81.35°W) at 1200 UTC 9 April and 1200 UTC 10 April 2020 (not shown) indicated a strengthening subsidence inversion at low levels, with increased drying throughout the 900–300 hPa layer. Both air parcel trajectories remained far enough above the surface to avoid humidification as they traveled over the Gulf of Mexico. In contrast, the 0000 UTC and 1200 UTC air parcels arriving at the 850 hPa level (Figure 10a,b, respectively) underneath the eastern dry tongue contained low-density humid air and spent enough time at or near the surface to become humidified from their initial dry conditions, reaching a relative humidity of 90 and 80% just after landfall, respectively. The 850 hPa air parcel that arrived in the region at 0000 UTC 13 April (Figure 10a) was part of the converging air streams of water vapor at low levels described in Part I (cf., Figure 12d of Part I) originating over the sub-tropical Atlantic Ocean (Figure 5b).

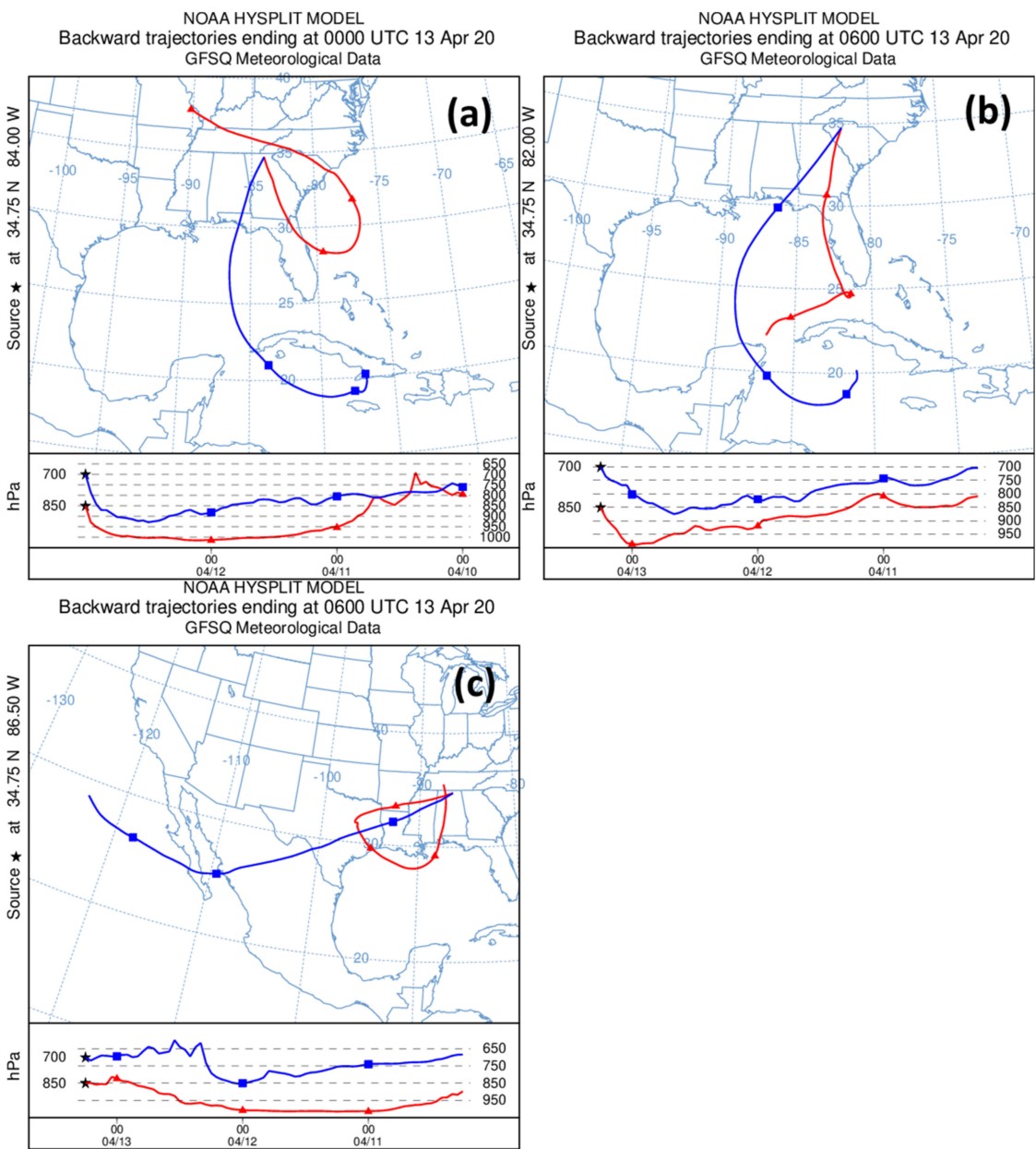

**Figure 10.** HYSPLIT 72-h trajectories derived from GFS 0.25° analyses for air parcels ending at the 700 {blue} and 850 {red} hPa levels at (**a**) 0000 UTC and 34.75°N; 84°W, (**b**) 0600 UTC and 34.75°N; 82°W, and (**c**) 0600 UTC 13 April 2020 and 34.75°N; 86.5°W.

Air parcel 72-h trajectories ending at the 700 and 850 hPa level for a location underneath the sub-tropical jet core are displayed in Figure 10c (end point; 34.75°N, 86.5°W at 0600 UTC). The air parcel trajectory at the ~700 hPa level ending in the western dry tongue (Figure 7b) originated over the eastern Pacific Ocean, gradually subsiding within the sub-tropics, until making landfall along the west coast of Mexico. Once over land, the air parcel experienced ascent as it was ingested into a position downstream of the large-scale upper-air cyclone. The air parcel traveled at mid-levels throughout its journey, remaining well below saturation, with a relative humidity of only 30% as it entered northern Louisiana. In contrast, the air parcel ending at the 850 hPa level underneath the western dry tongue spent at least 24 h (0000 UTC 11 April–0000 UTC 12 April 2020) being humidified by the warm underlying waters of the northern Gulf of Mexico, reaching a relative humidity of 87% as it made landfall near the surface. For each of the 700 hPa level 72-h air parcel trajectories described above in the discussion of Figure 10, the late period

of their journey was undoubtedly spent experiencing wet-bulb cooling as precipitation of the April storm fell through their position. Hence, the relatively low-$\theta e$ layer of air at ~700 hPa in Figure 9 was the combination of a very dry air source at the parcel origin and evaporative cooling once the air parcel reached the precipitation shield of the April storm.

### 3.1. Rainfall Evolution

#### 3.1.1. Rain Gauge Observations

Hourly rain rate and time (RRt) profiles of both heavy rainfall events in early 2020 (Figures 11 and 12) were created to allow comparison of the macroscopic cause and trigger precipitation structures that may have been unique to each event. Only selected RRt profiles are shown in Figures 11 and 12 as there is much similarity in profile patterns between numerous gauges. Reasons for inclusion of particular rain gauge profiles in the figures are unique to each event and will be explained in the description that follows.

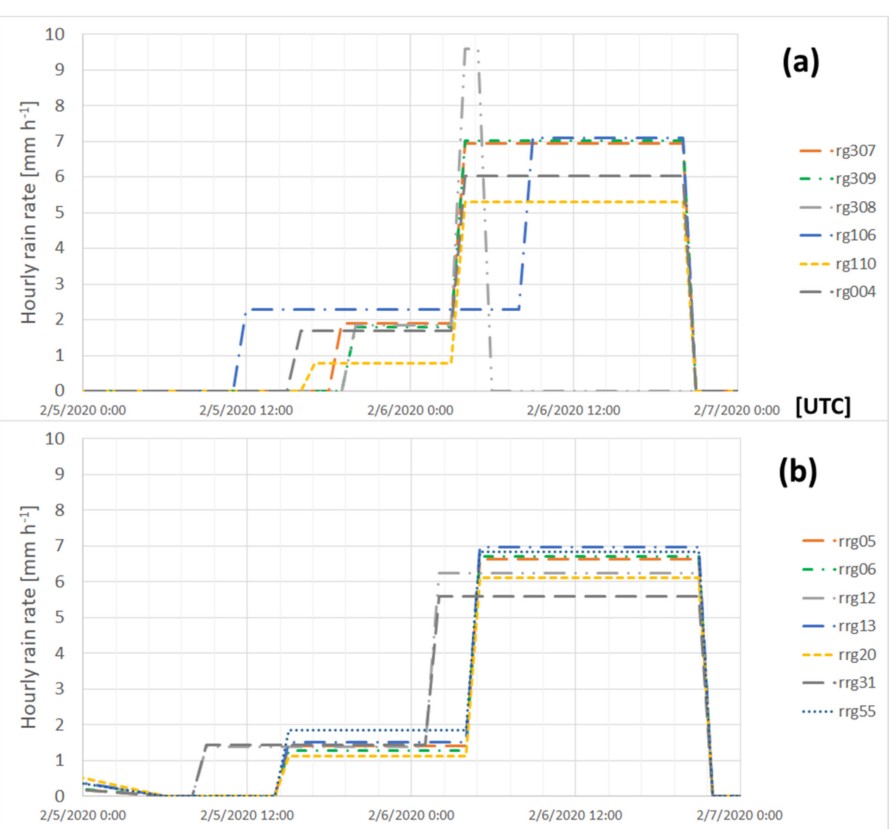

**Figure 11.** Hourly rain rate and time (RRt) profiles of the February 2020 event over the period 0000 UTC 5 February–0000 UTC 7 February 2020 for selected rain gauge observations of the (**a**) PRB and (**b**) CRB.

The Bunches Bald landslide occurred during or slightly after passage of the February 2020 event (inset of Figure 1) and was located 5.1 km southwest of rain gauge #110 and 14.7 km south of rain gauge #307 of the Duke GSMRGN (Figure 11a). For comparison, RRt profiles of the highest event accumulation gauges (#308 and #309) of the February event are included in Figure 11, along with RRt profiles at a climatologically rainy location in the PRB (#106) and the second highest elevation gauge (#004) of the gauge network, located in the southernmost portion of the PRB. The RRt profile of the closest rain gauge to Bunches Bald (#110) was unspectacular and, potentially, its suppressed cause and trigger phase mean rain rates the result of downsloping as it is located near Garretts Gap, at a lower elevation than Bunches Bald (1672 m). A review of animations of radar reflectivity observed by the WSR88D station of the Morristown, TN National Weather Service (NWS) office (KMRX, not shown) indicated cloud elements moving from the southwest, orthogonal to the ridgeline

containing Bunches Bald, confirming likely downsloping at gauge #110. The RRt profile at the next closest rain gauge to Bunches Bald (#307) was not noteworthy compared to the gauge of the greatest event accumulation (179 mm, #309). The ordinary nature of the gauge #307 RRt profile was further confirmed when compared to those of seven selected rain gauges in the CRB (Figure 11b), showing similar cause phase hourly mean rain rates (1.9 mm h$^{-1}$) and trigger phase hourly mean rain rates (6.9 mm h$^{-1}$). Recall that no landslides occurred during or after the February event in close proximity to the CRB. In hindsight, the lack of high-impact downstream flooding observed between Newport and Chattanooga, Tennessee (Part I) after the February event suggested the pre-event deep soil moisture was low (storage well below capacity) such that a minimal number of landslides (two) were triggered by a long duration and high accumulation event.

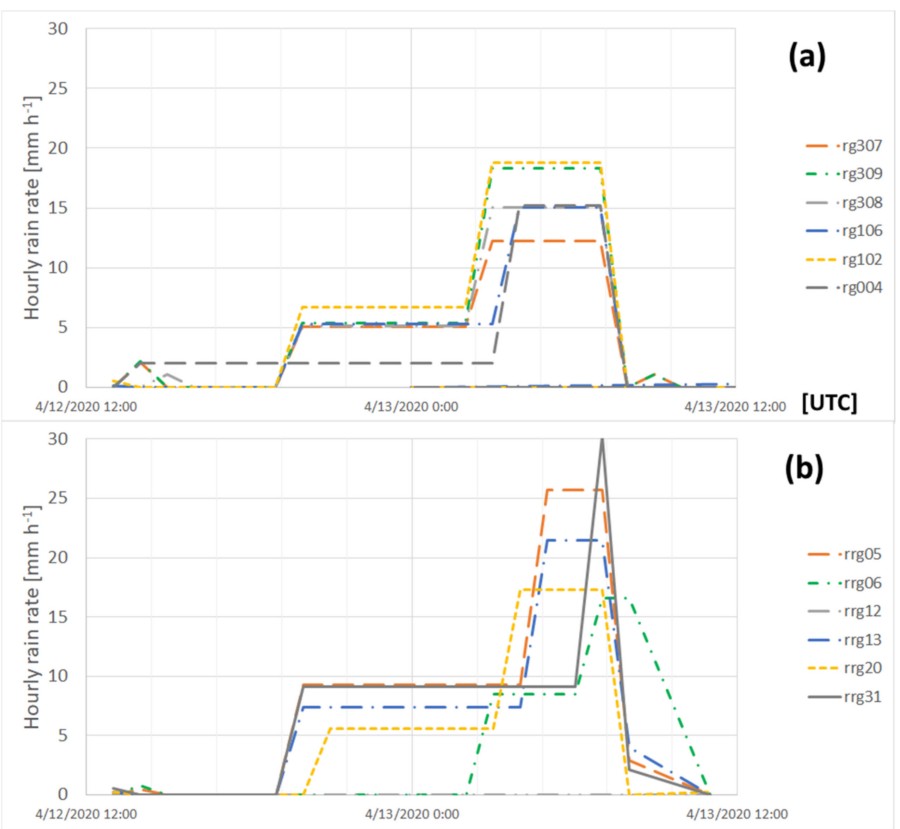

**Figure 12.** As in Figure 11, except for the April 2020 event covering the period 1200 UTC 12 April–1200 UTC 13 April 2020 for selected rain gauge observations of the (**a**) PRB and (**b**) CRB. Note the range of hourly rain rate axis is three times greater than that of Figure 11.

Consistent with the hypotheses and findings of Collins et al. [13], the lack of organized MCEs embedded in the middle sector of the February event having high-intensity rain rates prevented widespread landslide occurrence in the southern Appalachians. A supposition, consistent with the findings of Nippgen et al. [3] and Part I of the study, is the February event added significantly to the deep volumetric water storage of the southern Appalachians, predisposing the region to landslides when the second heavy rainfall event entered the southern Appalachians in April.

In contrast, the April event consisted of numerous sizable embedded MCEs (observed by the WSR88D station at the Greer, SC NWS office (KGSP, not shown)), particularly near the shared border between North and South Carolina, which gave hourly mean rain rates several times greater in the PRB (Figure 12a) and CRB (Figure 12b) than observed during the February event. Most of the 21 landslides triggered during or just after the April event occurred north and within 30 km of the CRB (Figure 1). The gauge observing the highest event accumulation in the PRB (#102, Figure 12a) had hourly mean rain rates during

the cause phase matching the rates observed during the trigger phase of the February event (~7 mm h$^{-1}$). Although its trigger phase hourly mean rain rate was nearly triple (~19 mm h$^{-1}$) that of the February event, its magnitude and duration (5-h) was insufficient to trigger a landslide in its proximity or for any gauge location in the PRB. This would suggest a combination of an insufficient preconditioning of the deep soil water storage of the PRB during the February event (mean event accumulation of 138 mm) and/or an insufficient accumulation during the cause and trigger phase of the April event (mean event accumulation of 94 mm) for exceeding the deep soil storage capacity of the PRB.

In the CRB to the south, the preconditioning of the deep soil water storage during the February event (mean event accumulation of 157 mm) and/or a sufficient accumulation during the cause and trigger phase of the April event (mean event accumulation of 126 mm) exceeded the deep soil storage capacity of the CRB, resulting in widespread landslides in the region. Three gauges in the CRB observing the largest trigger phase hourly mean rain rate and duration (#05, #13, and #20, Figure 12b) are located along the ridgeline of its northernmost border, on south-facing slopes, ideal positioning for upslope flow during the second-half of the middle storm sector during the April event. The RRt profile of gauge #31, located at the southernmost ridgeline of the CRB boundary, observed a one-hour trigger phase having an hourly mean rain rate of 30 mm h$^{-1}$, the consequence of a linear MCE (Figure 13) located at the leading edge of the 700 hPa level western dry tongue (Figures 7b and 9b).

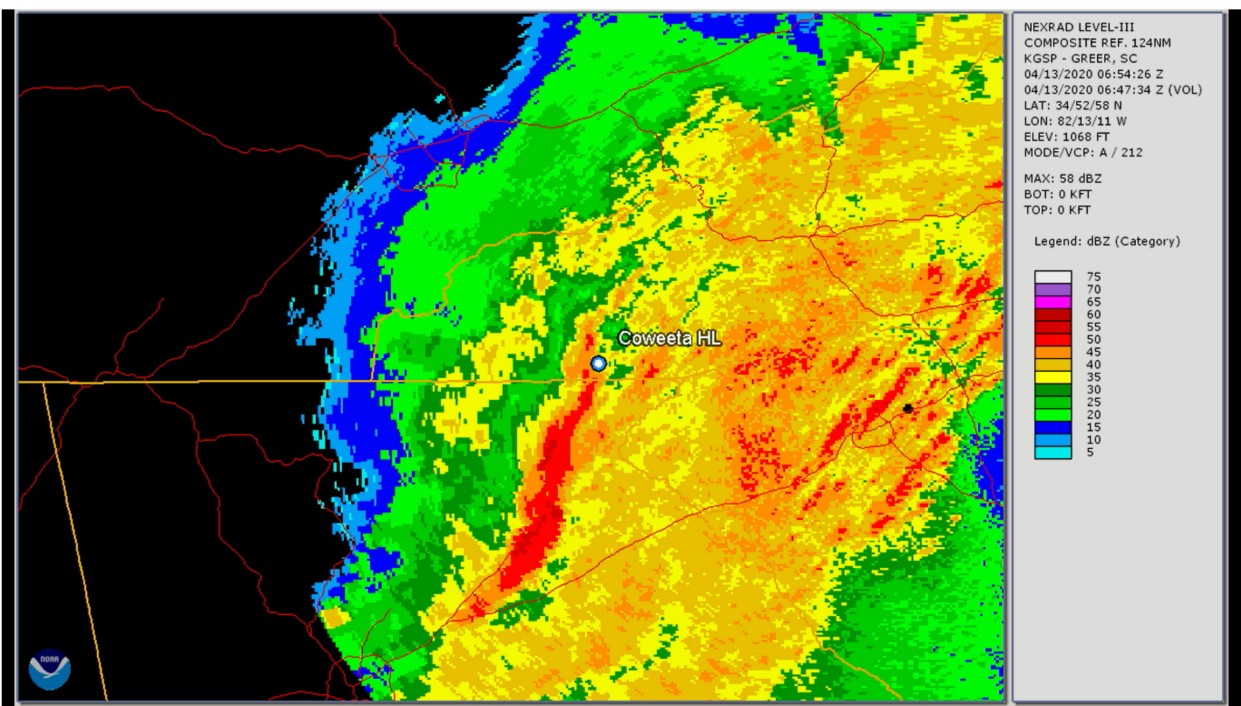

**Figure 13.** High-resolution composite reflectivity observations of the KGSP (Greer, SC) NWS WSR88D radar valid 6:54 UTC 13 April 2020. Location of the CRB is highlighted by the marker labeled "Coweeta HL".

### 3.1.2. Space-Borne Rainfall Estimates

A comparison of CMORPH-based QPEs averaged over the $1° \times 1°$ landslide focus region (Figure 4d) to the advective timescale-averaged 15-min "instantaneous" rain rate observations of the Duke GSMRGN are provided in Figure 14. As described above and in Part I, the February 2020 event (Figure 14a,c,e) was of longer duration and had the higher event accumulation compared to the April 2020 event (Figure 14b,d,f). CMORPH accumulation estimates suggest the February was nearly double that of the April event (Figure 14a,b). The time of greatest CMORPH-based mean and maximum rain rates in both events occurred when the integrated vapor transport (IVT) plume associated with the

AR was centered over southwestern North Carolina (cf., Figure 6a,b of Part I). Observed elevated mean and maximum rain rates of the Duke GSMRGN corresponded to the passage of ARs of both events, but also showed significant rates at earlier stages of the February 2020 event, associated with the smaller and lesser-organized MCEs described above (1200 UTC 5 February and 0500 UTC 6 February 2020; Figure 14c,e). CMORPH-based mean and maximum rain rate estimates during the early stages of the April event AR passage (0000–0500 UTC 13 April; Figure 14d,f) were nearly half the amounts observed by the Duke GSMRGN.

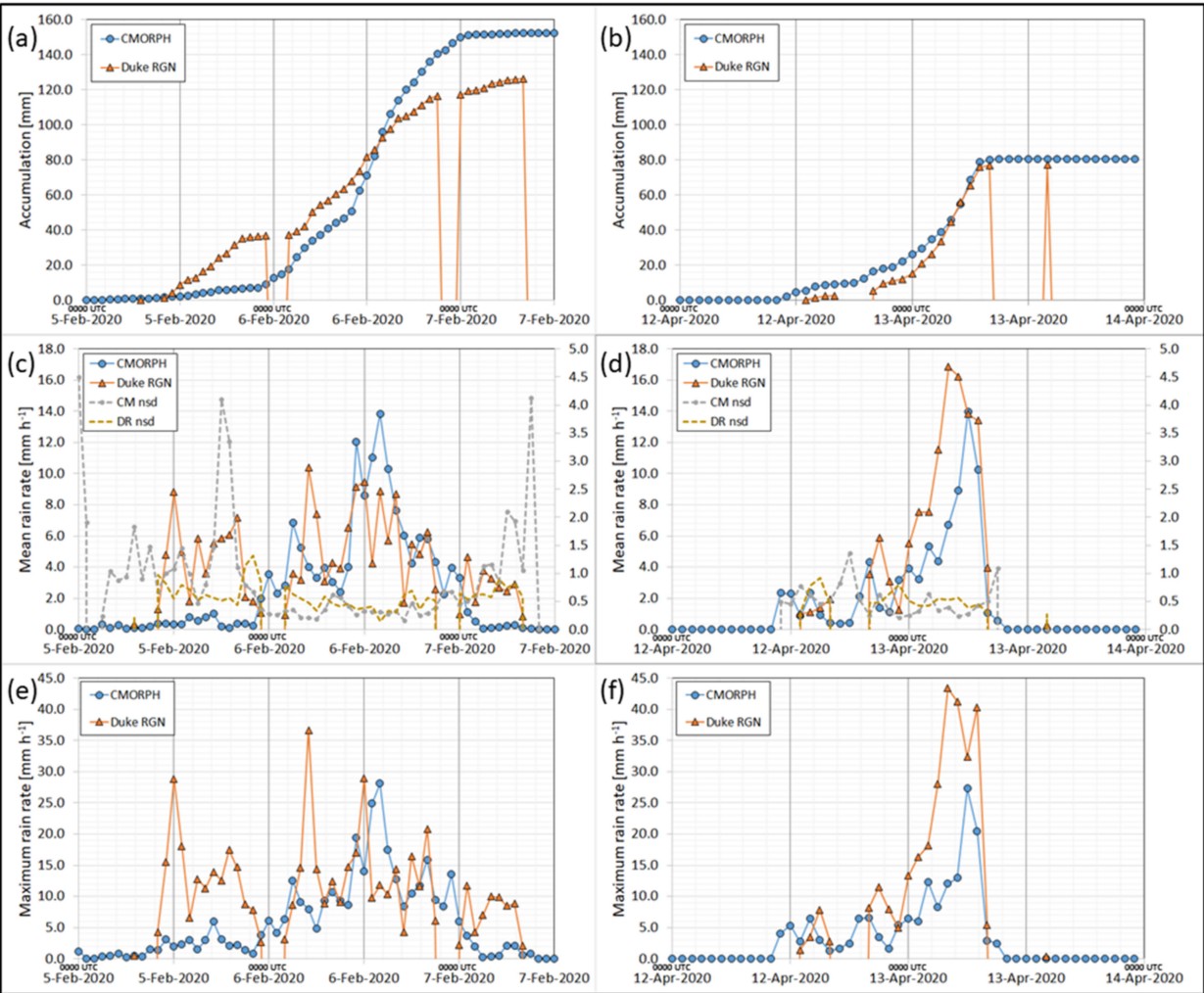

**Figure 14.** CMORPH-based event accumulation (mm) for the (**a**) February and (**b**) April 2020 storm, mean rain rate (mm h$^{-1}$) over 1° × 1° landslide focus region (Figure 4d) for the (**c**) February and (**d**) April 2020 event, and maximum rain rate (mm h$^{-1}$) over the 1° × 1° landslide focus region for (**e**) February and (**f**) April 2020 event highlighted in the blue-circle curve. Time series of time-averaged rainfall observations of the Duke GSMRGN are included in the orange-triangle curve for comparison. Panels (c) and (**d**) also contain normalized rain rate standard deviations for CMORPH (grey-circle dashed curve) and Duke GSMRGN observations (gold dashed curve). Time resolution of the plots is every hour. Drops in the Duke GSMRGN curves of panels (**a**–**f**) represent averaging periods when no tips were recorded at any of the rain gauges in the PRB.

A comparison of Enterprise-based QPEs averaged over the 1° × 1° landslide focus region to the advective timescale-averaged 15-min "instantaneous" rain rate observations of the Duke GSMRGN are provided in Figure 15. The lower observed Duke GSMRGN accumulation totals of the February event in Figure 15a are due to the "late" start of the Enterprise record of observations, which initiate at 0000 UTC 6 February 2020. In spite of the late start of the data record, the overall Enterprise-based total event accumulation of

the February storm is comparable to the total CMORPH estimate (Figure 14a). This "catch up" in accumulation was made possible by a particularly active period (1000–1600 UTC 6 February 2020; Figure 15c,e) assessed by the Enterprise algorithm in which mean and maximum rain rates topped out at 19.4 and 48.4 mm $h^{-1}$, respectively, in the landslide focus region. Elevated mean rain rates were also noted over the region during this period by CMORPH (Figure 14c), which corresponded to the middle and late stages of an AR passing over the focus region (Figure 6). In contrast, observed mean and maximum rain rates of the Duke GSMRGN over the same period did not exceed 10 and 30 mm $h^{-1}$, respectively (Figure 15c,e). Unfortunately, MiRS and GPROF QPEs were unavailable within the active AR period during the February event (Table 1) due to their orbital geometry. Average accumulation observed by the CHLRGN in the CRB over this 6-h period was 49.6 mm, compared to the Enterprise estimate of 65.2 mm. Although physically plausible, the high Enterprise QPEs over the 6-h period due to the relatively small and transient convective elements of the February event (Figure 6c), compared to the larger and slower moving elements of the April event (Figures 7c and 13), raise suspicions. Enterprise QPEs during AR passage of the April event (Figure 15d,f) over the landslide focus region were much reduced in mean and maximum rain rate compared to the February event. The GPROF 53.28 mm $h^{-1}$ maximum rain rate estimate at 0740 UTC 13 April 2020 (Table 1) was a reflection of the reinvigorated convective line depicted in Figure 13 as it exited the mountains and moved east of the landslide focus region. The MiRS QPEs of 0659 UTC 13 April 2020 (Table 1) were valid five minutes after the time of the convective line radar image depicted in Figure 13 as it was crossing the southern Appalachians. Closer inspection of the reduced Enterprise QPEs during AR passage of the April event revealed warming cloud tops in GOES-16 IR imagery that forced a change in rain-type classifications relating IR window brightness temperatures to surface rain rate. Hence, the reduced Enterprise QPEs during AR passage of the April event were likely in error, rather than the elevated estimates during AR passage of the February event.

Given the plausibility of the Enterprise QPEs during AR passage of the February event, it is worth considering that the nearest gauges of the Duke GSMRGN to the initiation point of the landslide at Bunches Bald were too far north and east to be representative of atmospheric conditions at the bald. In other words, the observed maximum rain rates at the Duke GSMRGN gauges #110 and #307 during the 6-h period (18.1 mm $h^{-1}$; 1000–1600 UTC 6 February 2020, Figure 11a), 5.1 km northeast and 14.7 km north of the Bunches Bald ridgeline, respectively, were outside the influence of a small-scale and transient MCE passing overhead, having a localized rain rate as high as 48.4 mm $h^{-1}$. The third closest of the Duke GSMRGN gauges (#106) to Bunches Bald, located 17.0 km to the southeast, observed a maximum rain rate during the 6-h period of 29.0 mm $h^{-1}$. Total Enterprise event accumulation estimates of the February storm (not shown) increased southward, toward the Blue Ridge Escarpment. Hence, Enterprise QPEs during the February event raise the possibility of the cause and trigger by atmospheric processes being sufficient to initiate the landslide at Bunches Bald. A survey of the landslide zone near Bunches Bald by the NCGS showed it was initiated on a human-modified slope, near where a previous heavy rainfall event had initiated a landslide in April 2019 [41]. Wooten et al. [11] and others have shown that landslides can be initiated with lower intensity peak rain rates where the surface soil has been modified by human activity. The increased vulnerability of the soil at this location likely reduced the critical cause and trigger landslide thresholds, resulting in an isolated landslide for relatively modest rain rates.

**Table 1.** Rain rate estimates of the MiRS and GPROF (v2017) algorithms over the 1° × 1° landslide focus region (Figure 4d) compared to time-averaged 15 min rain rate observations of the Duke GSMRGN (final three columns) for the February (upper eight rows) and April (lower two rows) 2020 events.

| Time/Date | QPE Algo. | Mean (mm h$^{-1}$) | Max. (mm h$^{-1}$) | Stand. Dev. (mm h$^{-1}$) | Mean (mm h$^{-1}$) | Max. (mm h$^{-1}$) | Stand. Dev. (mm h$^{-1}$) |
|---|---|---|---|---|---|---|---|
| 18:43 UTC 5 Feb 2020 | MiRS | 0.31 | 2.20 | 0.54 | 3.57 | 10.22 | 1.91 |
| 19:12 UTC 5 Feb 2020 | GPROF | 0.05 | 1.23 | 1.62 | 6.82 | 17.16 | 2.48 |
| 07:06 UTC 6 Feb 2020 | MiRS | 2.43 | 4.60 | 0.91 | 2.78 | 6.59 | 1.32 |
| 07:10 UTC 6 Feb 2020 | GPROF | 2.76 | 7.97 | 1.62 | 2.33 | 4.86 | 0.89 |
| 18:16 UTC 6 Feb 2020 | GPROF | 4.49 | 10.50 | 2.39 | 5.18 | 14.65 | 4.11 |
| 18:25 UTC 6 Feb 2020 | MiRS | 3.11 | 6.50 | 1.02 | 3.58 | 12.90 | 2.27 |
| 06:47 UTC 7 Feb 2020 | MiRS | 0.11 | 0.60 | 0.00 | 2.49 | 8.50 | 2.10 |
| 07:54 UTC 7 Feb 2020 | GPROF | 0.04 | 0.73 | 0.20 | 0.66 | 1.31 | 0.36 |
| 06:59 UTC 13 Apr 2020 | MiRS | 8.48 | 17.20 | 1.17 | 11.81 | 25.48 | 4.57 |
| 07:40 UTC 13 Apr 2020 | GPROF | 6.27 | 53.28 | 5.03 | 14.18 | 35.64 | 7.97 |

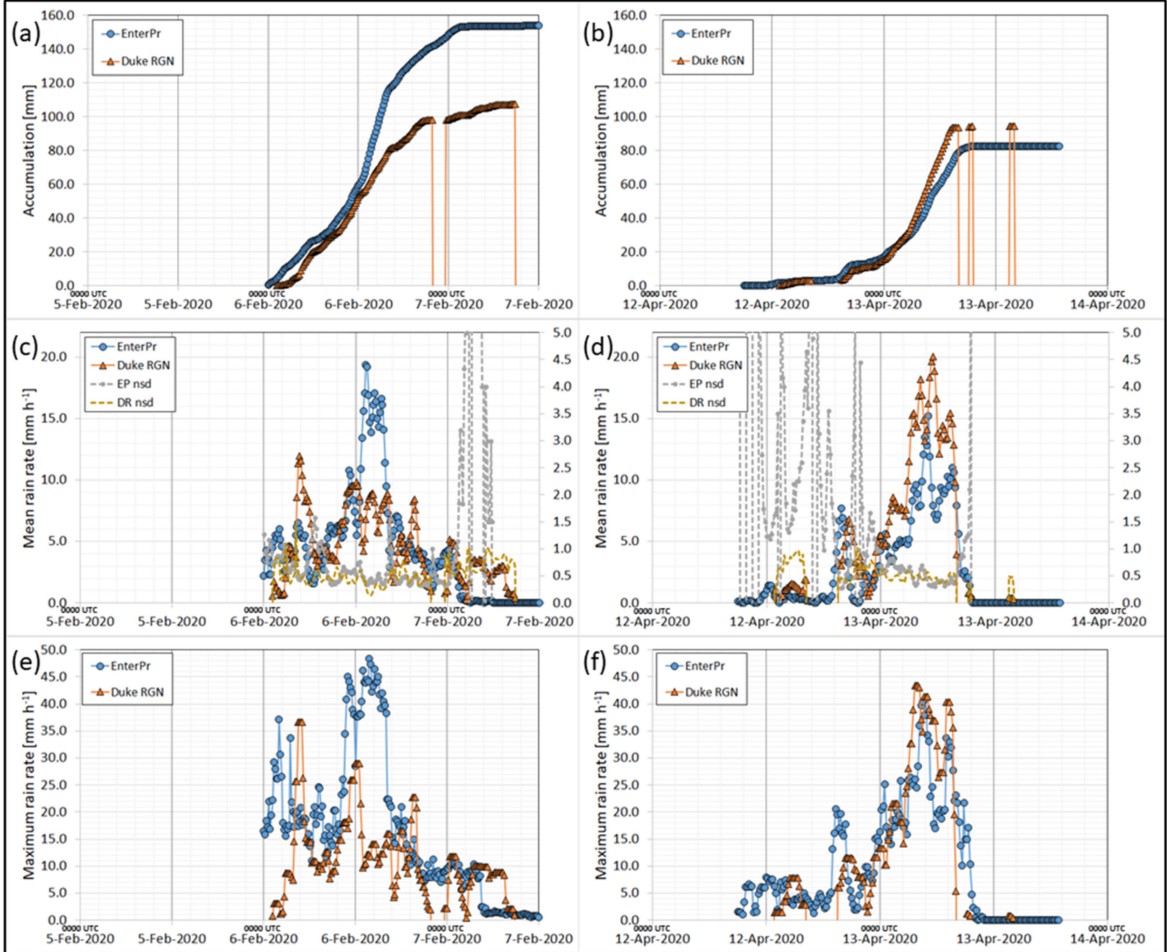

**Figure 15.** As in Figure 14, except based on Enterprise accumulation estimates (mm) for the (**a**) February and (**b**) April 2020 storm, mean rain rate (mm h$^{-1}$) over 1° × 1° landslide focus region (Figure 4d) for the (**c**) February and (**d**) April 2020 event, and maximum rain rate (mm h$^{-1}$) over the 1° × 1° landslide focus region for (**e**) February and (**f**) April 2020 event highlighted in the blue-circle curve. Time series of time-averaged rainfall observations of the Duke GSMRGN are included in the orange-triangle curve for comparison. Panels (**c**) and (**d**) also contain normalized rain rate standard deviations for Enterprise (grey-circle dashed curve) and Duke GSMRGN observations (gold dashed curve). Time resolution of the plots is every 10 min. Drops in the Duke GSMRGN curves of panels (**a**–**f**) represent averaging periods when no tips were recorded at any of the rain gauges in the PRB.

### 3.2. Other Space-Borne Nowcasting Aids

Vertical profiling capabilities of instrumentation aboard the polar-orbiting NOAA-20/ATMS and S-NPP/ATMS satellites offer unique insights into the thermodynamics of these extratropical storms. Retrieved profiles of temperature and water vapor allow the analysis of convective instability potential through vertical cross section plots of $\theta$e (Figure 16). After passage of the AR during the February event (1825 UTC 6 February 2020, Figure 16a), a relatively stable environment was evident through all layers of the atmosphere as $\theta$e clearly increased with height above the surface, consistent with the general pattern of the GFS analysed vertical sections (Figure 8). The transition from maritime tropical to continental polar air masses was evident on the left-hand side of the section at 1825 UTC 6 February with lower $\theta$e values moving in from the west. As addressed in the discussion of observed rain rates shown in Table 1, no polar-orbiting observations of $\theta$e were available near the time of the AR being centered on the landslide focus study area (1200 UTC 6 February 2020, Figure 4d). Hence, analysis of the evolution of pre-, concurrent-, and post-AR environmental stability as estimated by MiRS was not possible for the February heavy rainfall event.

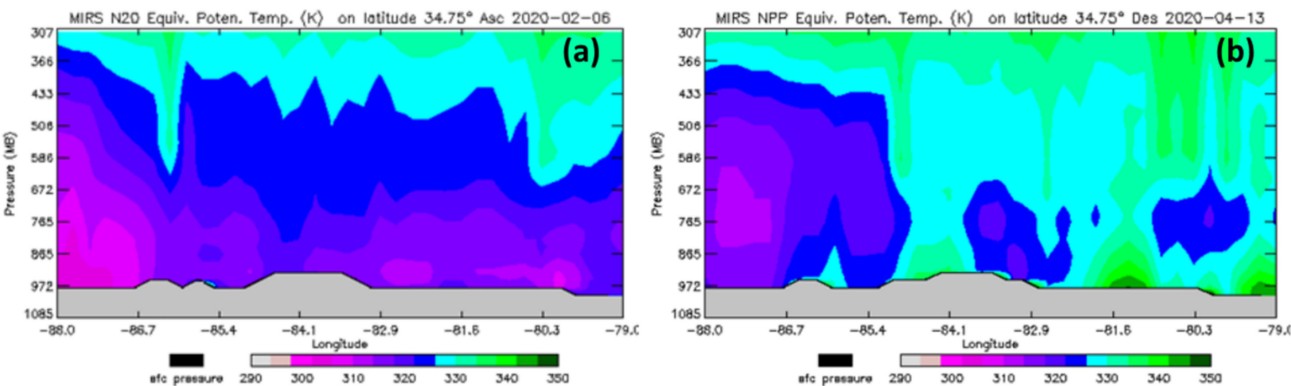

**Figure 16.** Vertical cross-section (endpoints at 34.75°N; 88°W (L), 79°W (R)) of MiRS-estimated equivalent potential temperature retrieved during overpasses by (**a**) NOAA-20/ATMS at 1825 UTC 6 February 2020 and (**b**) S-NPP/ATMS at 0659 UTC 13 April 2020.

In contrast, passage of the AR during the April 2020 event (Figure 16b) showed isolated "pockets" of instability just below the 700 hPa level, similar to the GFS-analysed vertical sections (Figure 9), as $\theta$e decreased with height from the ground to 700 hPa. The low-stability pocket in the middle of the section (Figure 16b) is likely associated with the western "dry tongue" (source region over the eastern Pacific Ocean) while the pocket near the eastern edge of the section was associated with the eastern "dry tongue" (source region located south of Cuba). The low-$\theta$e air moving in from the west (left-hand edge of Figure 16b) represented the transition to dryer continental polar air as maritime tropical air and its parent cyclone moved eastward at 0659 UTC 13 April 2020.

### 4. Discussion

The lack of mid- and lower-layer soil moisture measurements makes the prediction of landslide initiation in the mountains challenging. As suggested by Nippgen et al. [3], watershed memory implies that runoff and storage within the watershed requires a long period view (six months or less) of precipitation events affecting it, beyond the 30-day lag period investigated in Miller et al. [4]. Although absolute water storage of mountainous watersheds is likely to be an unknown in the foreseeable future, remote sensing observations of downstream flooding (e.g., VIIRS/ABI algorithm of Part I), post-event upper-soil layer moisture drying rate (e.g., SMOPS; cf., Figure 10 of Part I), and geostationary QPEs (e.g., CMORPH and Enterprise algorithms) allow indirect estimates in the trends of relative water storage in the watersheds.

Figure 17 is the bi-monthly integrated watershed-averaged precipitation accumulation climatological anomaly of the PRB (blue) and CRB (orange) with an early September 2019 starting point. Time series in Figure 17 are calculated from the observed and climatological time series in Figure 13 of Part I. Bi-monthly data points (first half; days 1–15, second half; days 16-end of month) are plotted on the 7th and 22nd day of each month and represent the summed bi-monthly anomalies up to the point. For example, the CRB anomaly point plotted on 22 December 2019 represents the summed CRB-averaged precipitation accumulation anomaly over the period 1 September–15 December 2019. No matter the slope of the lines in Figure 17, precipitation is always adding water to the watershed that is either stored or lost through runoff and evapotranspiration. The stronger the positive slope, water input via precipitation is exceeding watershed loss to a greater degree, and storage is increasing at a faster rate. With a large enough archive of geostationary QPEs over each mountainous watershed, the curves of Figure 17 could be recreated to serve as input to relative water storage assessments, negating the need to have rain gauge networks covering the entire mountainous region. Post-event observed downstream flooding and upper-soil layer moisture drying, coupled with observed precipitation event accumulation and duration, would also contribute as input to the qualitative estimate of how close watershed storage might be to its capacity or limit. Atmospheric model predictions (forecasting) or observations of "instantaneous" rain rates by geostationary or polar-orbiting QPEs (e.g., GPROF, MiRS algorithms) during passage of each individual extratropical storm (nowcasting) during its cause and trigger phases can, with the relative water storage estimate of a mountainous watershed as described above, give a landslide forecaster the sense of whether landslides within a watershed are likely or unlikely.

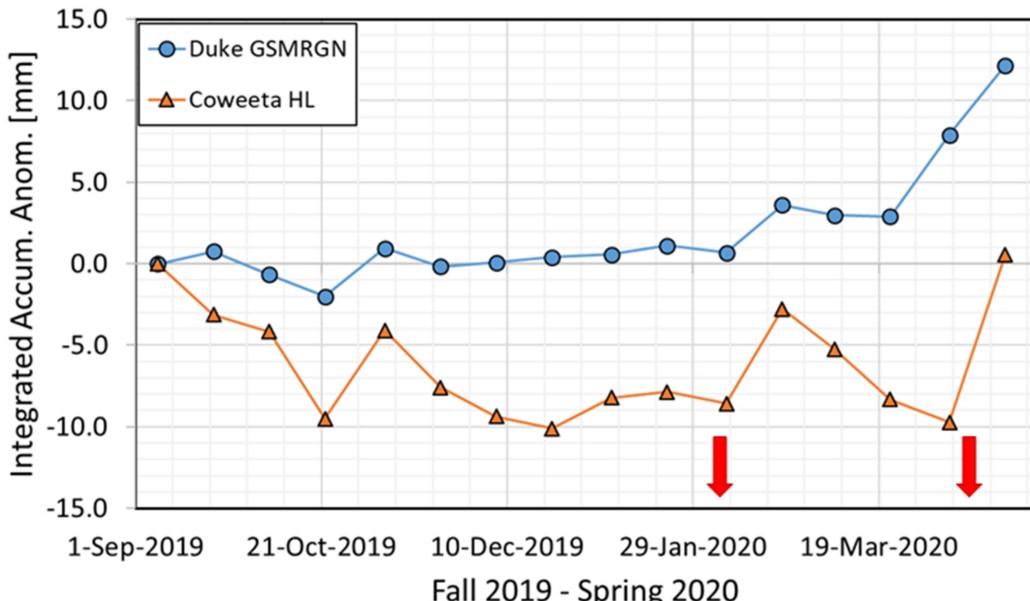

**Figure 17.** Bi-monthly integrated watershed-averaged precipitation accumulation climatological anomaly for fall 2019–spring 2020 of the PRB (11-year climatology; blue) and CRB (86-year climatology; orange) with an early September 2019 starting point. Time series are calculated from the observed and climatology time series in Figure 13 of Part I. Focus of this study is on the two events (5–7 February and 12–13 April 2020) highlighted in the diagram with a red arrow.

Using 10 April 2020 as an example, a forecaster looking at Figure 17 can see that a significant input of water to the PRB and CRB watersheds occurred between the first and second half of February 2020 (primarily during the 5–7 February 2020 event), no more than 65 days in the past. Investigation of downstream flooding via VIIRS/ABI observations during February would show insignificant impacts, SMOPS would indicate rapid drying of the upper-soil layer, and geostationary QPEs of the 5–7 February 2020 event would flag

the event as a high accumulation and long duration event, its associated rainfall, thus, being able to infiltrate into mid and lower (deep) layers of each watershed, significantly increasing watershed storage. Additionally, the lack of widespread landslides after the February 2020 heavy rainfall event would serve as another clue that mid- and lower-soil layer water storage capacity had not been exceeded. The greater event accumulation over the CRB, along with the stronger positive slope of the CRB curve for February (Figure 17) would alert the forecaster to the higher likelihood of landslide initiation in close proximity to the CRB. Atmospheric model forecasts of the 12–13 April 2020 event, with sufficient skill, would further alert the forecaster to the likelihood of landslides triggered by the presence of MCEs in the middle sector (trigger phase) of the storm. "Sufficient skill" implies that atmospheric models would accurately simulate the observed "dry tongues" and AR, necessary ingredients for sustaining the landslide-triggering MCEs of the April 2020 event. Vertical sections of MiRS-estimated $\theta$e (Figure 16) nicely highlighted differences in environmental stability between the two heavy rainfall events. Such a product could prove useful as a nowcasting tool in evaluating convective instability potential during the passage of ARs, if imagery was available more frequently than the local times of ascending and descending nodes of polar-orbiting satellites.

It is possible widespread landslides may have occurred had the February 2020 event occurred, instead, in November or December 2019. Post-Olga, at the end of October 2019, produced two shallow landslides (rupture depths of ~0.91 m, [15]) in the southern Appalachians without widespread flooding, indicative of Olga rainfall serving as a replenishing event for mid- and lower-soil layer water. In reality, the 98-day lag time between the passage of tropical cyclone Olga and 5 February 2020 allowed ample time for runoff to reduce the mid- and lower-soil layer moisture content in the southern Appalachians. Despite sporadic rainfall events between the end of October 2019 and 5 February 2020, soil water replenishment in the interim period (Figure 17) was too weak to keep soil water content near capacity. As a result, the "instantaneous" (15-min) rate rates (Figure 14c,e and Figure 15c,e) and cause and trigger phase hourly rain rates and duration (Figure 11) of the 5–7 February 2020 event proved insufficient for overwhelming soil moisture storage capacity. Only two superficial landslides occurred after the February 2020 event, indicating its associated rainfall served primarily to replenish the mid- and lower-soil layer water 65 days before the April 2020 heavy rainfall event.

Once a sufficient number of rainfall case studies has been collected over the southern Appalachians to establish a meaningful precipitation climatology, a future study can begin to investigate positive slope thresholds of the curves in Figure 17 that could form the basis of landslide "watch" issuances for relevant basins. In addition to defining positive slope thresholds, examination of corresponding space-based QPEs, downstream flooding and upper-layer soil moisture drying estimates, along with observed (NCGS) landslides of each event will help improve understanding of lag thresholds between a significant replenishing event and an upcoming rainfall event. The lag threshold will undoubtedly also be a function of the predicted "instantaneous" (15-min) rate rates and cause and trigger phase hourly rain rates and duration of the upcoming rainfall event.

## 5. Conclusions

In conclusion, a comparison of two heavy rainfall events of early 2020 in which the long duration event (February) yielded two documented landslides, while the shorter duration event (April) yielded 21 landslides spread over a broad area near the CRB in the southern Appalachian mountains, illuminated key factors resulting in the different outcomes. The large-scale upper-level trough/cyclone supporting each storm was initiated beforehand by anticyclonic Rossby wave breaking (Part I). A significant distinction between the two events in the large-scale weather pattern was the role of the downstream sub-tropical anticyclone. The anticyclone of the 12–13 April 2020 event was responsible for a humid airstream originating from the sub-tropical Atlantic Ocean converging with another airstream having origins in the Caribbean Sea/ Gulf of Mexico (Figure 5b of this paper and Figure 11b,d of

Part I). The downstream anticyclone also provided a source region of dry air above the PBL south of Cuba. Because of its strengthening subsidence, the dry air stream guided poleward by the sub-tropical anticyclone and ingested into the April storm, immediately above the moisture-laden AR, was a sustained dry air source (Figure 10a,b), the eastern "dry tongue". The omnipresent dry layer yielded long-term available potential energy that, combined with synoptic-scale lift, released convective instability (Figures 9 and 16b) and long-lived and relatively large MCEs.

The largest and longest lived of the MCEs in the April event was found on the western flank of the AR (Figure 9b), in a position one might expect to find a narrow cold frontal rainband (Figure 13) corresponding to the second-half of the middle storm sector stage (Figure 2b), during the trigger phase. The source of the western "dry tongue" air stream was a sub-tropical anticyclone in the eastern Pacific Ocean (Figure 10c) and was ingested into the large-scale upper-air cyclone as it moved poleward. The dry airstream, in a storm-relative sense, moved eastward faster than the progression of the AR so that the airstream overtook it and provided yet another environment of sustained convective instability (Figures 9b and 16b) during passage of the late middle storm sector.

Conditioning of the mid- and lower-soil layer moisture by the 5–7 February 2020 event 65 days before the April 2020 event was a key ingredient to the widespread landslide response observed near the CRB during and after the rainfall of the 12–13 April storm. This conclusion was based on space-borne observations showing a lack of downstream flooding and a rapid drying of the upper-soil layer after the February 2020 (Part I). These observations, coupled with the scant number of observed landslides, led to the conclusion that the February 2020 event was primarily a soil moisture-replenishing event.

A significant number of landslides that occurred during and after the April 2020 storm were concentrated near the CRB (Figure 1). The storm ranked as an extreme (top 2.5%) rainfall event in both the CRB and PRB over the 86- and 11-year data archive, respectively. A natural question worthy of attention is why landslides did not occur near the PRB during or after 13 April 2020. The event accumulation mean rainfall of the February 2020 event in the PRB was 138 mm (compared to 157 mm of the CRB, Part I), and it could be argued that the antecedent soil moisture was not as well conditioned for landslides to occur in the PRB in April. However, it is unlikely that an event accumulation mean rainfall difference of 19 mm is significant enough to be a satisfactory answer to the question. Additionally, rainfall during 5–7 February was sufficient for triggering the isolated landslide near Bunches Bald. The most likely explanation for differences in the post-April rainfall event landslide response near the two basins is related to differences in localized storm structure during passage of the April storm, resulting in diminished cause and trigger phase hourly rain rates in the PRB (Figure 12). The PRB is located a fair distance north and west of the BRE. As stated previously, orography of the BRE provides additional lift beyond the dynamics of the storm that leads to generally heavier rainfall for basins located closer to it. Additionally, the horizontal expanse of the eastern and western dry tongues diminished northward (Figures 5b and 7b) and triggered MCEs near the PRB were of smaller horizontal extent and lifespans. Finally, local orography lining the southern boundary of the PRB has been shown to diminish rainfall accumulation in the PRB for events having a low-level southerly wind [4]. Wind directions at the 700 hPa level (at the center of the cross sections of Figure 9) just before and during the time of AR passage in the April event had a significant southerly component (~205°).

**Author Contributions:** Conceptualization, D.M. and R.F.; methodology, D.M., M.A., R.F., C.G., B.K., S.L., V.P., S.W., and P.X.; formal analysis, D.M., M.A., C.G., B.K., S.L., V.P., S.W., and P.X.; data curation, D.M., C.G., B.K., V.P., and P.X.; writing—original draft preparation, D.M.; writing—review and editing, D.M., R.F., and V.P. All authors have read and agreed to the published version of the manuscript.

**Funding:** Maintenance costs of the Duke GSMRGN were supported by NOAA through the Cooperative Institute for Satellite Earth System Studies under Cooperative Agreement NA19NES4320002. NASA grants NNX07AK40G, NNX10AH66G, and NNX13AH39G covered installation and original maintenance costs of the Duke GSMRGN to Ana Barros at Duke University.

**Institutional Review Board Statement:** Not applicable.

**Informed Consent Statement:** Not applicable.

**Data Availability Statement:** Data sets utilized in this study not having an appropriate URL listing in the manuscript are available upon request to the lead author.

**Acknowledgments:** We gratefully acknowledge the NOAA GOES-R and JPSS programs for providing support to the satellite algorithms and the Duke GSMRGN. The authors are grateful for the support and assistance of Paul Super and Tom Remaley of the National Park Service, land owners who have permitted the installation of a rain gauge on their property, University of North Carolina at Asheville and Duke University students, and Kyle, Don, Hugh, and Roger, support personnel of the Waynesville Watershed. We are grateful for the numerous discussions with Rick Wooten and Corey Scheip of the North Carolina Geological Survey. We also acknowledge the collection and sharing of rainfall observations by A. Christopher Oishi and Patsy Clinton of the USDA Forest Service, Coweeta Hydrologic Laboratory contained in their data archive. We also gratefully acknowledge the helpful comments of several anonymous reviewers who greatly improved the quality of the manuscript. A review by Sheldon Kusselson also helped to improve the manuscript. Color scales used in figures with shading were designed by Cynthia Brewer and Mark Harrower (http://colorbrewer2.org, accessed on 12 December 2020). Additionally, appreciation of Shawn Milrad (http://www.shawnmilrad.com/gempak-color-tables-and-stuff, accessed on 12 December 2020) must be noted whose instructions allowed the pleasing color tables of "colorbrewer" to be incorporated into the shading of GEMPAK-generated figures.

**Conflicts of Interest:** The manuscript contents are solely the opinions of the authors and do not constitute a statement of policy, decision, or position on behalf of NOAA or the U. S. Government. The authors declare no conflict of interest. The funders had no role in the design of the study; in the collection, analyses, or interpretation of data; in the writing of the manuscript, or in the decision to publish the results.

## Appendix A

**Table A1.** Location and elevation of the 32 tipping bucket rain gauges comprising the Duke GSMRGN and of the nine NOAH IV weighing rain gauges (WRGs) comprising the CHLRGN.

| Duke GSMRGN Gauge Attributes | | | | | | | | CHLRGN Gauge Attributes | | | |
|---|---|---|---|---|---|---|---|---|---|---|---|
| Gauge | Lat. | Lon. | Elev. (m) | Gauge | Lat. | Lon. | Elev. (m) | Gauge | Lat. | Lon. | Elev. (m) |
| RG002 | 35°25.5′ | 82°58.2′ | 1731 | RG109 | 35°29.7′ | 83°02.4′ | 1500 | WRG06 | 35°3.62′ | 83°25.8′ | 687 |
| RG003 | 35°23.0′ | 82°54.9′ | 1609 | RG110 | 35°32.8′ | 83°08.8′ | 1563 | WRG05 | 35°3.63′ | 83°27.9′ | 1144 |
| RG004 | 35°22.0′ | 82°59.4′ | 1922 | RG111 | 35°43.7′ | 82°56.8′ | 1394 | WRG20 | 35°3.89′ | 83°26.5′ | 740 |
| RG005 | 35°24.5′ | 82°57.8′ | 1520 | RG112 | 35°45.0′ | 82°57.8′ | 1184 | WRG31 | 35°1.96′ | 83°28.1′ | 1366 |
| RG008 | 35°22.9′ | 82°58.4′ | 1737 | RG300 | 35°43.5′ | 83°13.0′ | 1558 | WRG13 | 35°3.75′ | 83°27.4′ | 961 |
| RG010 | 35°27.3′ | 82°56.8′ | 1478 | RG301 | 35°42.3′ | 83°15.3′ | 2003 | WRG41 | 35°3.32′ | 83°25.7′ | 776 |
| RG011 | 35°23.7′ | 82°54.9′ | 1244 | RG302 | 35°43.2′ | 83°14.8′ | 1860 | WRG12 | 35°2.84′ | 83°27.5′ | 1001 |
| RG100 | 35°35.1′ | 83°04.3′ | 1495 | RG303 | 35°45.7′ | 83°09.7′ | 1490 | WRG55 | 35°2.39′ | 83°27.3′ | 1035 |
| RG101 | 35°34.5′ | 83°05.2′ | 1520 | RG304 | 35°40.2′ | 83°10.9′ | 1820 | WRG96 | 35°2.72′ | 83°26.2′ | 894 |
| RG102 | 35°33.8′ | 83°06.2′ | 1635 | RG305 | 35°41.4′ | 83°07.9′ | 1630 | | | | |
| RG103 | 35°33.2′ | 83°07.0′ | 1688 | RG306 | 35°44.7′ | 83°10.2′ | 1536 | | | | |
| RG104 | 35°33.2′ | 83°05.2′ | 1587 | RG307 | 35°39.0′ | 83°11.9′ | 1624 | | | | |
| RG105 | 35°38.0′ | 83°02.4′ | 1345 | RG308 | 35°43.8′ | 83°10.9′ | 1471 | | | | |
| RG106 | 35°25.9′ | 83°01.7′ | 1210 | RG309 | 35°40.9′ | 83°09.0′ | 1604 | | | | |
| RG107 | 35°34.0′ | 82°54.4′ | 1359 | RG310 | 35°42.1′ | 83°07.3′ | 1756 | | | | |
| RG108 | 35°33.2′ | 82°59.3′ | 1277 | RG311 | 35°45.9′ | 83°08.4′ | 1036 | | | | |

## Appendix B

**Table A2.** List of uncommon abbreviations used in describing the study.

| Abbreviation | Definition |
| --- | --- |
| AR | Atmospheric River |
| BRE | Blue Ridge Escarpment |
| CHLRGN | Coweeta Hydrologic Laboratory Rain Gauge Network |
| CRB | Coweeta River sub-Basin |
| Duke GSMRGN | Duke Great Smoky Mountains Rain Gauge Network |
| MCE | Mesoscale Convective Element |
| PRB | Pigeon River Basin |
| RRt | Hourly Rain Rate and time |
| ULTRB | Upper Little Tennessee River Basin |

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
