# Peer review of "A Study of Two Impactful Heavy Rainfall Events in the Southern Appalachian Mountains during Early 2020, Part II; Regional Overview, Rainfall Evolution, and Satellite QPE Utility"

_remotesensing, doi:10.3390/rs13132500_

Round 1

Reviewer 1 Report

  1. I didn’t find the line numbers in this manuscript.
  2. Page 8 Results: “The cause phase of the February 2020 event was unlike the schematic of Fig. 2b in that the southern Appalachian Mountains were entirely on the warm-air side of the frontal zone located in eastern Tennessee and northern Alabama (Fig. 3a) in the early sector of the storm.” Maybe the schematic of Fig. 2b could be modified for different events.
  3. Page 8: How does strong warm air advection and overrunning along the warm front (Figs.5a and 5b) contribute to sporadic precipitation? Is this a key reason that the cause phase of the April 2020 event (Fig. 3b) more closely resembled the early sector Fig. 2b schematic?
  4. Page 10: The breadth of the stronger winds in the lower layer at 1200 UTC 6 February (Fig. 8b) was a response to the strengthening baroclinic zone evident below the 700 hPa level. How does it contribute to the enhanced integrated vapor transport within the AR?
  5. Page 15: It pointed out that the RRt profiles of the highest event accumulation gauges of the February events were #308 and #309. What are the hourly mean rain rates at cause phase and trigger phase for #308 and #309, respectively? Are they similar to each other?
  6. Page 21: The April event was of relatively short duration (30-h); while the February event was a particularly long duration event (54-h). What is the effect of rainfall event duration on landslide based on this study? It is valuable to explain clearly.

Author Response

Please find my response to your comments in the attached document.

Reviewer 2 Report

For landslide hazard analysis the main factors usually involved are: slope, lithology, soil moisture, precipitation etc. Only precipitation is taken into account in the present study, which is not sufficient for a proper analysis. Precipitation and the soil moisture are connected, despite the lag time between the occurrence of precipitation and the maximum water content in the soil. Increased temperature and wind diminish the upper soil humidity, but after a rain event or snowmelt process the soil moisture in the profile is high for a long period because the intermediate flow and percolation take a long time, due to low water velocity in porous media.

The rain duration and rain intensity series influences directly the volumetric water content. At the same time, the soil moisture is not completely reflected by the wind speed (figure 9),  only the upper soil humidity being directly influenced by the wind.

I suspect the event in February, characterized by a high accumulation and a long duration supplied consistently the soil moisture (but not enough to generate the landslide). Due to long duration of precipitation, the water had enough time to infiltrate into deeper layers, contributing to the moisture increase. The soil was probably dry and absorbed most of the precipitation; the proof is that no significant flood was recorded downstream in February. When the April rains occurred the soil moisture was quite high (except probably the upper soil due to the wind), and the new precipitation contributed to the increase of soil humidity to the limit that the sliding forces exceeded the stability forces. At the same time, although the total rain depth in April was about half of the accumulation in February, “extended downstream flooding between Newton and Chattanooga, Tennessee” occurred. This is an additional proof that the soil humidity was at high values, leading to a  high runoff coefficient.

I think the two events (February and April) cannot be separated and should be considered together for landslides analysis.

Page 16

I express my reserves related to the following phrase:

“The observed cause (trigger) phase hourly mean rain rate of 9 and 7 mm h-1 (26 and 21 mm h-1) at gauges #05 and #13, respectively, and 9-h (3-h) duration can serve as estimates of necessary meteorological thresholds for triggering landslides, given the relatively dry antecedent soil moisture conditions leading up to the 12 April 2020”

Page 22:

The same reserves related to the next sentence:

In other words, did uncommon atmospheric constituents come together at the right time and in the proper amounts to force landslides, independent of the condition of the underlying soil?

Author Response

(The authors gave the same response as above.)

Reviewer 3 Report

The present paper describes the relationship between regional meteorological condition analyzed using satellite QPE and landslides in a mountainous area. The paper describes the case study on two heavy rainfall events in a mountainous region. Although the validity of the authors' methods has not yet been tested, authors describe methods and results in detail. Thus, the present paper has possibility to provide important information on the risk prediction of landslides in the study area. Authors are suggested to make revision focusing on the effectiveness of the authors' methods as well as the necessity of further analysis. Especially the conclusion section is rather confusing. I suggest authors to describe compactly focusing on how the authors method is applicable for prediction of landslides without including information on geological condition of the study area.

The followings are minor comments.

  1. Abbreviation list will be helpful.

  1. Figure 2: Explanation on the difference of gray tone should be included in the figure legend..

  1. Figure 10: Explanation on the difference of colored lines should be included in the figure legend..

Author Response

Please find my response to your comments in the attached file.

Round 2

Reviewer 2 Report

No further comments or suggestions.

Author Response

Please find my response to reviewer comments of version 2 of the manuscript in the attached file.

Reviewer 3 Report

The revised manuscript has been improved. Authors added one figure in the conclusion section; however, the figure should be moved to the results section and discussion on the figure should be described in the discussion section.

Author Response

(The authors gave the same response as above.)
